# Scaling Online Complex Event Detection with Synthetic Supervision and Mamba-Based Neural Algorithmic Reasoning

## Abstract

Modern machine learning models excel at detecting individual actions, sounds, or scene attributes from short, localized observations. However, many real-world tasks, such as in smart cities and healthcare, require reasoning over high-level *complex events* (*CE*s): spatiotemporal, rule-governed patterns of short-term *atomic events* (*AE*s). Complex event detection (CED) is challenging due to long temporal dependencies, generalization beyond the training horizon, sparse *CE*-level supervision, and cognitively demanding annotation, as *CE* labels often depend on ordering, duration, negation, and completion-time semantics. These challenges are further amplified in an online setting that requires causal, streaming inference with limited computation. We identify the primary bottleneck in online CED as learning robust *CE* rules, and propose a Neural Algorithmic Reasoning framework that decouples rule learning from low-level sensor semantics by (i) generating large-scale synthetic *AE*-level concept traces to pretrain a Mamba-based *CE*-rule reasoner, and (ii) introducing an adapter that learns to map raw sensor inputs into the reasoner's latent space using limited, labeled sensor data. We introduce a controlled online multilabel CED testbed built from real-world multimodal sensor clips and rule-generated *CE* labels, with stress-test settings that vary sensor noise, distribution shift, and the window size used to segment streaming sensor sequences. Experiments show that our method matches or exceeds the strongest baselines under these stress tests and longer-horizon generalization, with $5\times$ fewer labeled sensor sequences and $10$–$20\times$ fewer FLOPs. Code and dataset available at: /r/naroce-C7F6/.

## 1 Introduction

Recent advances in machine learning have greatly improved low-level sensory perception tasks such as action recognition and sound event detection. However, many real-world intelligent systems must go beyond recognizing short-lived patterns in sensor streams: they need to *reason over high-level event patterns* that unfold over extended temporal (and often spatial) horizons and reflect meaningful scenarios or critical changes in the underlying state of the world.

This capability is commonly formalized as **complex event detection (CED)** (Cugola & Margara, 2012; Xing et al., 2019; Roig Vilamala et al., 2023; Wang et al., 2024; Han et al., 2025). A **complex event (CE)** is a rule-governed pattern composed of multiple **atomic events (AEs)**. *AE*s are low-level semantic events that can be inferred from short, fixed-duration perception windows (typically seconds). In contrast, *CE*s capture higher-level (spatio)temporal relationships among *AE*s, such as ordering, duration, repetition, temporal gaps, overlap, and negation, and may span minutes or hours. For example, in a restaurant, a *CE* such as a *sanitary protocol violation* may be defined as an employee using the restroom and subsequently touching kitchen surfaces without an intervening hand-washing action. In a smart-city setting, a *CE* such as a *coordinated security threat* may be inferred from distributed observations, such as two baggage drop-offs at different locations separated by tens of minutes, followed by a vehicle pick-up captured by another camera. In both cases, the target *CE* cannot be inferred from any single observation; it emerges from the temporal composition and ordering of multiple *AE*s over long horizons. Another defining characteristic of CED is

that annotations are typically provided **only for high-level *CE* occurrences**, while fine-grained temporal annotations of the underlying *AE*s are not available due to the labor-intensive annotation burden.

Detecting *CE*s from noisy sensory data poses several fundamental challenges. First, a model needs to robustly identify a small number of key *AE*s while ***ignoring large amounts of irrelevant activity***, often formalized as "don't-care" elements (denoted as $X$). For instance, a sanitary protocol may be expressed as "Use restroom" $\rightarrow X \rightarrow$ "Wash hands" $\rightarrow X \rightarrow$ "Touch kitchen surfaces", where $X$ may include unrelated actions such as walking or sitting. Allowing such flexibility dramatically enlarges the space of acceptable sequences, especially as temporal horizons grow.

Second, *CE*s exhibit ***variable and long-range temporal dependencies***. The time gap between two critical *AE*s may range from seconds to hours, yet the semantic rule defining the *CE* remains unchanged. As a result, learning-based systems must extrapolate *CE* patterns beyond the fixed-length traces observed during training and remain robust to temporal variability at inference time, which is an inherently difficult requirement for models trained on finite data traces.

Third, many application domains demand ***online detection***. In scenarios such as eldercare, workplace safety, or building security monitoring, a *CE* must be recognized as it unfolds rather than through offline post hoc analysis. This requires maintaining a faithful internal state that summarizes relevant historical context and updating it continuously as new sensory observations arrive.

Beyond modeling challenges, ***dataset scale and supervision present a major bottleneck***. As a representative example, prior work (Han et al., 2025) reports learning-based methods trained on the order of 10,000 labeled 5-minute *CE* sequences, amounting to over 800 hours of sensor recordings, or approximately $4.8 \times 10^{10}$ raw samples for audio at $16\,\mathrm{kHz}$. Annotating a *CE* label is also cognitively demanding, requiring reasoning over long, structured event sequences spanning minutes or hours. Moreover, because many sensor streams (e.g., IMU or audio) are not directly human-interpretable, large-scale annotation is difficult without strict collection protocols or auxiliary modalities such as vision. Even the most relevant existing real-world event datasets (Oh et al., 2011) typically target loosely structured activities that do not require explicit long-horizon rule-based reasoning, and often contain only tens of instances per event. As a result, collecting realistic, large-scale *CE* data is prohibitive. This motivates two complementary goals pursued in this work: developing a methodology to reduce reliance on labeled sensor sequences, and building a controlled, scalable evaluation testbed that preserves key ingredients for online CED reasoning—real sensor observations, long-horizon temporal composition, and diverse *CE* types—while systematically varying sensor noise and distribution shift.

We propose `NAROCE`, a **N**eural **A**lgorithmic **R**easoning framework for data-efficient, robust, and scalable **O**nline **C**omplex **E**vent Detection. Our key idea is to decompose online CED into two subproblems: rule reasoning and sensor-to-reasoner alignment. Specifically, `NAROCE` contains (i) a reasoner, CE-NAR, which uses Mamba, a state-space model well suited for long-range dependencies, as its backbone and is pretrained on synthetic *AE*-level concept traces to learn *CE* rules independently of sensor data; and (ii) a lightweight adapter that maps sensor features to the latent reasoning space of CE-NAR. Synthetic *AE*-level concept traces can be generated by first-principles simulators or LLM-based synthesizers; here, we use LLMs to synthesize diverse simulator programs for trace generation.

To support systematic evaluation, we introduce DailyOCE, a controlled and scalable online CED testbed built from real-world multimodal sensor clips and rule-generated *CE* labels. By composing real sensor observations into long, structured event traces with diverse compositions, DailyOCE provides a practical proxy for training and evaluating online CED methods without requiring costly collection and manual annotation of real-world *CE* sequences. It captures key challenges of the target setting, including diverse and overlapping *CE* types, long-horizon temporal composition, sensor noise, distribution shift, and variation in the window size used to segment online streaming sensor data. It further enables evaluation of flexible-length, long-horizon rule extrapolation and cross-subject robustness. Across these settings, `NAROCE` matches or exceeds the strongest baselines on long-horizon generalization and robustness to sensor noise and shift, using only 20% as many labeled sensor sequences and 10–20× fewer FLOPs. These results highlight the value of pretraining on synthetic concept traces to learn robust *CE*-rule priors.

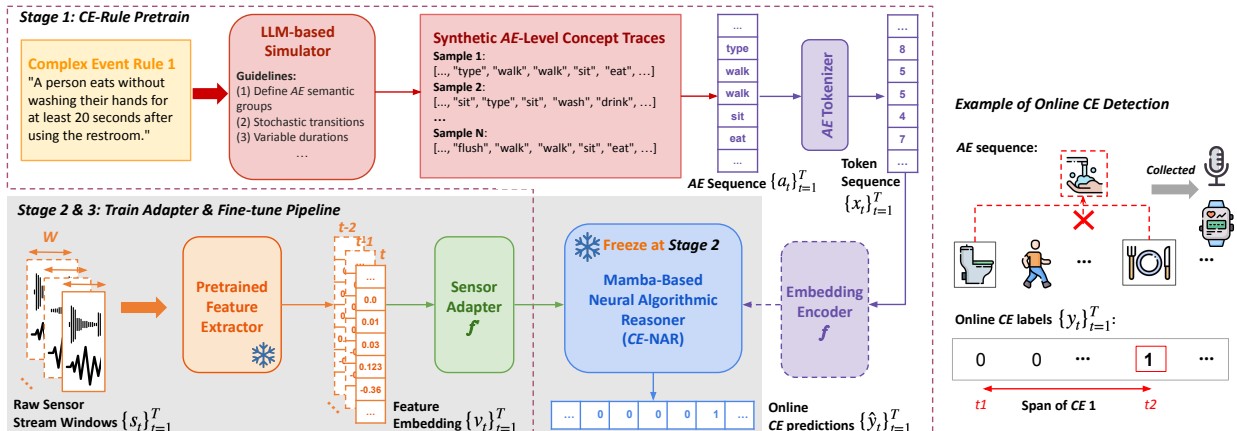

Figure 1: **Overview of the NAROCE framework.** In **Stage 1** (dashed box), an **LLM-based simulator** generates synthetic concept-level *AE* traces based on predefined complex event rules. The traces are **tokenized** and used to train the **Mamba-based CE-NAR** with corresponding online *CE* targets. In **Stages 2–3** (gray-shaded region), we first freeze **CE-NAR** and train a small **Sensor Adapter** $f'$ to map sensor feature embeddings (from a feature extractor) into **CE-NAR**'s latent reasoning space, then unfreeze and fine-tune the **Adapter** and **CE-NAR** jointly to adapt to the target distribution. Right: an example of online *CE* detection and its temporal span.

## 2 Related Work

**Complex Event Detection (CED).** Traditional CED has been extensively studied in structured databases, where mature rule-based engines detect symbolic event patterns from clean and structured inputs (Schultz-Møller et al., 2009; Debar & Wespi, 2001; Cugola & Margara, 2012). Recent efforts extend CED to unstructured, high-dimensional sensor data. For example, Wang et al. (2024) extracts symbolic labels (e.g., object types, attributes, bounding boxes) from the video and passes them to hand-written rules, but this approach is brittle to perception errors. Neurosymbolic methods (Xing et al., 2020; Roig Vilamala et al., 2023) improve robustness by integrating soft neural predictions with differentiable logic frameworks (De Raedt et al., 2007; Manhaeve et al., 2018; 2019), but still rely on user-defined rules and scale poorly. Han et al. (2025) explores fully data-driven methods for learning *CE* rules, showing that SSMs (Gu et al., 2022) like Mamba (Gu & Dao, 2024; Dao & Gu, 2024) are well-suited. However, their reliance on massive labeled sensor data remains a major drawback.

**Neural Algorithmic Reasoners (NAR).** NAR models learn classical algorithms using neural networks, enabling generalization on structured reasoning tasks such as sorting, shortest path, and graph traversal (Veličković et al., 2022; Ibarz et al., 2022; Velickovic & Blundell, 2021). Recent work combines a pretrained graph neural network (GNN)-based NAR with transformers to extend algorithmic reasoning to language tasks (Bounsi et al., 2024). However, these methods typically operate on clean, structured inputs and do not directly address noisy sensory observations. Inspired by NAR, we view online *CE*-rule detection as an algorithmic procedure, and train an NAR on synthetic *AE*-level concept traces to learn these rules. We then bridge the gap to raw sensor data through a learned sensor adapter, thereby decoupling high-level rule reasoning from low-level sensor semantic learning to improve data efficiency and generalization.

**Online Action Recognition.** Tasks such as Online Action/Action Start Detection, Temporal Action Localization, and Segmentation focus on real-time recognition of frame-level actions in streaming videos. Recent work improves long-range modeling using advanced temporal architectures and memory modules (An et al., 2023; Wang et al., 2023; Cao et al., 2023; Zhao & Krähenbühl, 2022; Chen et al., 2022a; Xu et al., 2021b; Wang et al., 2021). However, these methods target *flat, low-level, temporally localized* activities (Fabian Caba Heilbron & Niebles, 2015; Jiang et al., 2014) rather than activities or events that require structured and rule-based reasoning over ordering and long temporal gaps. Our task addresses an *orthogonal and complementary* challenge: rule-based reasoning over long sequences of *AE*s under sparse *CE*-only supervision.

## 3 Problem Formalization

### 3.1 Complex Event Definitions

**Definition 3.1.** An *atomic event* ($AE$) is a short-horizon, low-level semantic event inferred from a fixed *local* perception window over sensor data. The window length is application-dependent (e.g., seconds for human activity; tens of milliseconds for speech/audio). Let $A = \{a_1, \ldots, a_n\}$ denote the set of $AE$ types.

**Definition 3.2.** A *complex event* ($CE$) is a high-level event defined by a temporal and/or logical rule over multiple $AE$s that may span long horizons. Let $E = \{e_1, \ldots, e_k\}$ denote the set of $CE$ types of interest.

**Definition 3.3.** Each $CE$ $e_i \in E$ is specified by a *pattern function* $R_i$ over a subset of relevant $AE$ types $A_i \subseteq A$. Let $z_t \in A$ denote the $AE$ observed at window $t$. To ignore irrelevant $AE$s, define a mapping $g_i : A \to A_i \cup \{X\}$ that maps any $a \in A_i$ to itself and any $a \notin A_i$ to a *don't-care* symbol $X$ (i.e., ignored by $R_i$).

Let $\tilde{z}_t = g_i(z_t)$ and $\tilde{z}_{1:t} = (\tilde{z}_1, \ldots, \tilde{z}_t) \in (A_i \cup \{X\})^*$. The pattern function evaluates the $AE$ trace online:

$$R_i(\tilde{z}_{1:t}) \in \{0, 1\}, \quad t = 1, 2, \ldots \tag{1}$$

where $R_i(\cdot) = 1$ indicates that the temporal/logical pattern for $e_i$ is satisfied by the observed prefix up to window $t$. The *occurrence time* $t_{e_i}$ is the smallest window index $t$ such that $R_i(\tilde{z}_{1:t}) = 1$, i.e., the exact window when the pattern becomes satisfied.

We follow the pattern taxonomy in prior work (Han et al., 2025) and consider four primary categories: *Sequential Ordering*, *Temporal Duration*, *Repetition*, and *Combination*. All patterns $R_i$ considered here are *finite-state* and can be represented as finite state machines (FSMs). Formal definitions and examples are provided in Table 6.

### 3.2 Online & Multilabel Complex Event Detection

Assume a system receives a multimodal sensor stream $\mathbf{X} = \{\mathbf{X}^{(m)}\}_{m \in M}$, where modality $m$ is sampled at rate $r_m$. The stream is processed in windows of duration $W$. At window $t$, the segment for modality $m$ is

$$\mathbf{D}_t^{(m)} = \mathbf{X}^{(m)}(t) \in \mathbb{R}^{(r_m \cdot W) \times d_m}, \tag{2}$$

where $d_m$ is the feature dimension of modality $m$. We use the same window boundaries for all modalities and denote the multimodal input at window $t$ as $\mathbf{D}_t = \{\mathbf{D}_t^{(m)}\}_{m \in M}$.

Prior work (Han et al., 2025) formulates detection as a *multiclass* problem with mutually exclusive labels per window, assigning a single $CE$ label only at the window where the corresponding pattern becomes satisfied; the rule then resets to enable detecting subsequent occurrences. However, in real-world scenarios, occurrences of different $CE$s can overlap in time and multiple $CE$ types may be detected in the same window; therefore, we extend the formulation to **online multilabel CED**. Accordingly, each window $t$ is labeled with a binary vector $\mathbf{y}_t \in \{0, 1\}^{|E|}$, where $\mathbf{y}_t[i] = 1$ indicates that $CE$ $e_i$ is detected at window $t$; multiple entries can be 1 in the same window, representing co-occurring $CE$s. We adopt the same satisfaction-time labeling semantics, i.e., if an $e_i$-pattern spans from window $t_1$ to $t_2$, then only $\mathbf{y}_{t_2}[i] = 1$ while $\mathbf{y}_{t_1:t_2-1}[i] = 0$, and $\mathbf{y}_{t_2+1}[i] = 0$ due to the reset for detecting subsequent occurrences. We refer to the interval $[t_1, t_2]$ as the **$CE$ temporal span**. As a result, online $CE$ labels are *temporally sparse and rare*.

The goal of an online CED model $f$ is to minimize the discrepancy between the predicted label $\hat{\mathbf{y}}_t$ and the ground-truth label $\mathbf{y}_t$ at each window using current and past inputs:

$$\min_f \ \ell(\hat{\mathbf{y}}_t, \mathbf{y}_t), \quad \text{with } \hat{\mathbf{y}}_t = f(\mathbf{D}_{1:t}), \quad t = 1, 2, \ldots \tag{3}$$

where $\ell(\cdot)$ is a multilabel classification loss.

Only coarse $CE$ labels $\mathbf{y}_t$ are provided for supervision. Fine-grained $AE$ annotations are not available, so the model must learn both $AE$-relevant semantics and long-horizon $CE$ rules under *distant & weak supervision* from nonlocal, sparse, and high-level $CE$ labels, in an *online multilabel* setting.

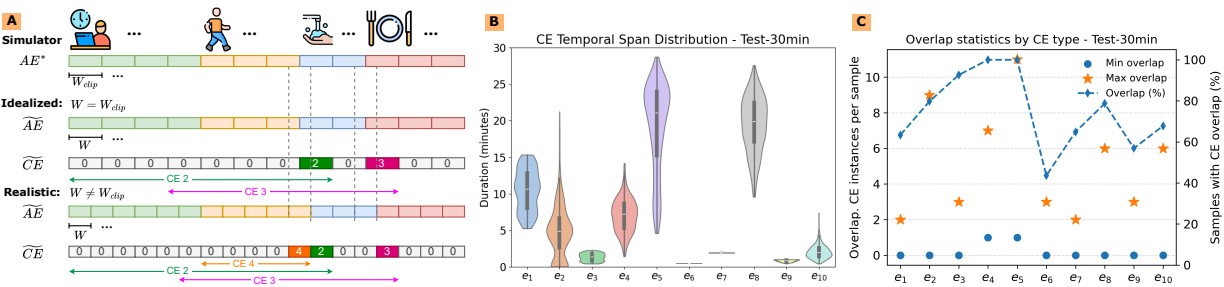

Figure 2: **DailyOCE visualization and analysis. (A)** The simulator generates sensor clips of $W_{\text{clip}}$ from the underlying $AE$ sequence, $AE^*$. Under idealized windowing, decision windows align with $AE$ segments, yielding noise-free window-level aggregated $AE$ labels, $\widetilde{AE}$, and derived $CE$ labels, $\widetilde{CE}$. Under realistic windowing, $AE$ transitions can occur within a window, introducing noise in $\widetilde{AE}$ and consequently $\widetilde{CE}$. Overlapping temporal spans of different $CE$ types are illustrated. **(B)** $CE$ temporal span distributions by $CE$ type in 30min test set. **(C)** Overlap statistics by $CE$ type: min/max overlapping $CE$ instances per sample (left) and % samples with overlap (right).

### 3.3 Online Inference Pipeline

A standard online CED pipeline (Han et al., 2025) first segments the continuous stream into non-overlapping decision windows of length $W$, yielding $\{s_t\}_{t=1}^{\infty}$. A **feature encoder** $m$ then extracts each (multimodal) sensor window $s_t$ into a latent embedding $v_t = m(s_t)$, producing an embedding stream $\{v_t\}_{t=1}^{\infty}$. Lastly, an **online complex event detector** $g$ takes the embedding history and outputs the prediction for window $t$:

$$\hat{\mathbf{y}}_t = g(v_{1:t}), \quad t = 1, 2, \dots \tag{4}$$

Our focus is designing $g$ to learn robust long-horizon $CE$ rules and sensor semantics from sparse $CE$-level supervision with limited labeled sensor data.

## 4 DailyOCE Benchmark

We build DailyOCE, a controlled multimodal testbed for online CED in a smart-health scenario, centered on human daily routines and protocols. DailyOCE is constructed from real-world multimodal sensor clips, while high-level $CE$ labels are generated from explicit temporal event definitions. This design enables systematic training and evaluation of long-horizon event reasoning under diverse, controlled conditions, without requiring costly collection and manual annotation of real-world $CE$ occurrences. For comparability with the only existing online CED benchmark we are aware of (Han et al., 2025), we retain the same ten $CE$ type definitions ($e_1$–$e_{10}$; Table 7). We then expand the benchmark along two design axes: (i) **multilabel $CE$ composition**, which allows co-occurring and overlapping $CE$s, and (ii) **windowing regimes**, spanning an *idealized* setting and a more challenging *perturbed* setting with imperfect temporal segmentation and observation noise.

### 4.1 Sensor Trace Generation and Labeling

We generate long multimodal traces of daily human routines by composing real-world IMU and audio clips under a stochastic activity simulator that mimics natural behavior: $AE$ durations are sampled from realistic ranges and $AE$-to-$AE$ transitions follow probabilistic patterns (details in Appendix C). We source IMU clips from WISDM (Weiss, 2019) and audio clips from ESC-70 (Moreaux et al., 2019), and we additionally collect in-house audio clips to cover all 9 $AE$ types. Each $AE$ instance is instantiated by sampling a $W_{\text{clip}} = 2.5\text{s}$ IMU/audio clip from its class and concatenating clips to form long sensor traces.

Online $CE$ labels are generated by applying an expert-designed FSM per $CE$ rule to the ground-truth $AE$ trace (example code in Appendix G). This simulator supports (i) *cross-subject* evaluation by generating test traces from held-out subjects and (ii) $CE$ rule generalization by varying $AE$ durations and inter-$AE$ gaps at test time while ensuring the target $CE$ patterns remain satisfied. We generate 5-minute traces for training, and 5/15/30-minute traces for evaluation.

**Axis I: Multilabel $CE$ composition.** Unlike Han et al. (2025), which uses mutually exclusive per-window labels and at most one active $CE$ per trace, DailyOCE allows temporal spans of different $CE$ classes to overlap in time. Consequently, multiple $CE$s may be detected in the same window, yielding multilabel supervision.

**Axis II: Windowing settings.** Fig. 2(A) illustrates how simulator-generated continuous sensor traces are segmented into online *decision windows* of length $W$ for labeling and evaluation under different windowing regimes. Note that training supervision remains *CE-only*; *AE* labels are used only inside the simulator to derive *CE* labels.

- *Idealized windowing.* The decision-window $W$ matches the underlying *AE* clip length ($W = 2.5s$), providing a controlled setting that removes window–*AE* length mismatch. We include: (i) *Aligned* windowing; (ii) *Phase-shifted* windowing, which keeps the same $W$ but offsets window start times by $\phi$ to test sensitivity to window-boundary placement; and (iii) *Adjacent-window fusion*, a controlled temporal-smoothing variant that mixes each window's feature embedding with the previous window using a weight $\lambda$.
- *Realistic windowing.* The decision-window $W$ differs from $W_{\text{clip}}$, reflecting reality where segmentation boundaries rarely match underlying *AE* boundaries. We first assign ground-truth *AE* labels at a finer resolution $\delta t$ (e.g., 0.5s), then derive each decision-window *AE* label by majority vote over its $\delta t$ segments; *CE* labels are computed by applying the *CE* FSMs to the resulting decision-window *AE* trace. This induces boundary mixing and label noise when a decision window overlaps multiple *AE*s. We evaluate multiple window sizes $W \neq W_{\text{clip}}$ to stress-test sensitivity to segmentation choices.

## 4.2 Dataset Realism and Diversity

The benchmark includes negative traces with no *CE* occurrence (default label $e_0$) to discourage false positives. Fig. 2(B) shows the diversity of temporal-span distribution of *CE* occurrences in the 30-minute test set, and (C) reports co-occurrence/overlap statistics across *CE* types, highlighting that multiple *CE*s can overlap within a trace, as in real-world scenarios. Additional analysis of *CE* distributions and overlaps is provided in Appendix H.

## 5 Methodology

Our design of `NAROCE` is guided by two hypotheses:

- **Hypothesis I:** *CE* rule reasoning is the primary bottleneck; decoupling rule reasoning from sensor semantics reduces reliance on expensive labeled sensor data.
- **Hypothesis II:** Pretraining on large-scale synthetic *AE*-level concept traces improves *CE* rule extrapolation to longer horizons and robustness under distribution shift.

Inspired by Neural Algorithmic Reasoning (NAR) (Veličković et al., 2022; Velickovic & Blundell, 2021; Ibarz et al., 2022), which uses neural networks to represent and learn algorithms from symbolic input-output pairs, we view each *CE* rule as an algorithm executed over an *AE* sequence. We use Mamba as the long-context reasoning backbone, CE-NAR, and train `NAROCE` in three stages, shown in Fig. 1:

- **Stage 1: Rule pretraining.** Train a Mamba-based CE-NAR on large-scale, low-cost synthetic *AE*-level concept traces to learn *CE* rules independently of sensor data.
- **Stage 2: Sensor adaptation.** With a small labeled sensor set, train the Sensor Adapter to map sensor embeddings into the latent reasoning space of CE-NAR.
- **Stage 3: Joint finetuning.** Unfreeze CE-NAR, and finetune it jointly with the Sensor Adapter for a few epochs using the same labeled sensor set.

## 5.1 Stage 1 - Training CE-NAR on Concept Traces

**Synthetic *AE*-level concept traces.** These text traces can be generated by any principled stochastic simulator. In this work, we design an LLM-based synthesizer that generates stochastic simulator programs, which are then used to produce concept traces at the decision-window granularity (one *AE* per window of length $W$) via structured prompting. Concretely, it follows a four-step procedure: (1) organizing *AE*s into *semantic groups* (e.g., hygiene, work); (2) defining *probabilistic transitions* between groups and between *AE*s within each group; (3) assigning *variable durations* to both groups and individual *AE*s; and (4) dynamically adjusting transition probabilities to increase the likelihood of the target *CE* occurring. For each *CE* rule, we use GPT-4o with temperature sampling set to 0.7 to generate 10 simulator programs, diversifying trace distributions and reducing bias. Appendix B provides full prompt details.

**Labeling.** To ensure accurate and consistent online *CE* labels, we deterministically label each concept trace by applying an expert-designed FSM per *CE* pattern (Section 4.1). LLMs are not used here to avoid hallucinations.

**Pretraining CE-NAR.** We pretrain CE-NAR as follows:

(1) Generate a large set of synthetic *AE*-level concept traces of length $T$ windows. Tokenize each trace $\{a_t\}_{t=1}^T$ with *AE* Tokenizer, which maps each *AE* concept $a_t$ to a discrete token $x_t$ via a fixed lookup vocabulary, producing $\{x_t\}_{t=1}^T$.

(2) Embed each token $x_t$ with a learnable embedding encoder, Embedding Encoder $f$, into a 128-dimensional latent space.

(3) Train Embedding Encoder $f$ jointly with CE-NAR on the concept trace dataset, then freeze CE-NAR for use as the *CE*-rule reasoner in later stages.

### 5.2 Stage 2 - Training the Sensor Adapter

We train the Sensor Adapter $f'$ to map the sensor embedding $v_t$ into CE-NAR's latent reasoning space. Here $v_t$ is a 128-dimensional embedding extracted from the multimodal sensor window by a Pretrained Feature Encoder.

(1) Remove $f$ from Stage 1 and introduce $f'$, which takes $v_t$ and outputs a 128-dimensional vector that serves as input to CE-NAR for window $t$.

(2) Keep CE-NAR frozen and train only $f'$ using labeled online *CE* sensor data.

We instantiate $f'$ as a small Mamba-based adapter to model short-range temporal structure within each window.

### 5.3 Stage 3 - Joint Finetuning

Finally, we unfreeze CE-NAR and finetune it jointly with the Sensor Adapter $f'$ for a few epochs using the same labeled sensor set. This stage provides light end-to-end adaptation, primarily refining the pretrained rule prior to the target *CE*-pattern distribution, while also allowing minor sensor-semantic adjustment beyond Stage 2.

**Inference.** At inference time, the pipeline only runs the Pretrained Feature Encoder, then the Sensor Adapter $f'$, followed by CE-NAR to produce online *CE* predictions.

### 5.4 Training Objective

Online *CE* supervision is extremely sparse (Section 3.2): in most windows, the *CE* label vector $\mathbf{y}_t \in \{0,1\}^{|E|}$ is all zeros, leading to severe class imbalance. We therefore adopt a *multilabel* version of Focal Loss to handle sparse supervision while supporting co-occurring *CE*s (Lin et al., 2017). Let $\hat{\mathbf{p}}_t \in (0,1)^{|E|}$ be the predicted probabilities. The loss over a dataset of $N$ sequences is

$$L_{\mathrm{FL}} = -\sum_{i=1}^N \sum_t \sum_{c=1}^{|E|} \ell_{\mathrm{FL}}\big(y_{i,t,c}, \hat{p}_{i,t,c}\big), \tag{5}$$

where the per-label loss for class $c$ at window $t$ is

$$\ell_{\mathrm{FL}}(y, \hat{p}) = \alpha\, y\, (1-\hat{p})^\gamma \log(\hat{p}) + (1-\alpha)(1-y)\, \hat{p}^\gamma \log(1-\hat{p}). \tag{6}$$

$\gamma$ is the focusing parameter that reduces the contribution of frequent classes, and $\alpha$ is the class weight. We use $\gamma = 2$ as recommended (Lin et al., 2017) and select $\alpha = 0.8$ for all positive *CE* classes via a small grid search. We apply this objective in all training stages of `NAROCE`.

## 6 Experiments

**Research Questions.** Our evaluation examines the two hypotheses introduced in Section 5 through four research questions. **RQ1. Data efficiency.** Does concept-trace pretraining reduce the amount of labeled sensor data required for online CED? **RQ2. Long-horizon generalization.** Does the pretrained *CE*

reasoner improve generalization to sequences longer than those observed during training, and how does this behavior scale with the amount of synthetic concept-trace pretraining? **RQ3. Robustness.** How robust is `NAROCE` to cross-subject distribution shift, sensor perturbations, window-boundary misalignment, and changes in decision-window size? **RQ4. Learned alignment.** Does the sensor adapter map sensor observations into a structured latent space that reflects the underlying *AE* semantics despite receiving only *CE*-level supervision?

**Datasets.** We *only* use 5-min labeled sensor data for training and validation, with a full budget of 10,000 training examples per idealized and realistic windowing setting in `DailyOCE`. `NAROCE` additionally pretrains on 40,000 5-min *AE*-level concept traces, which are easy and cheap to generate. For evaluation, the `DailyOCE` simulator generates test sets using held-out subjects: the 5-min test set measures cross-subject generalization, while the 15/30-min sets further increase *AE* durations and inter-*AE* gaps, requiring rule generalization beyond the training horizon.

**Pretrained Feature Encoder.** We use $m$ in Eq.4 as a pretrained, frozen encoder shared across all downstream models. We train $m$ on *AE* clips segmented with the same decision window $W$ used at inference and keep it fixed. Since this component is not the main focus of our work, we defer its architecture and training details to Appendix D.

**Baselines.** We compare against several complex event detector choices $g$ in Eq. 4. All baselines operate causally, using feature embedding extracted by $m$ up to window $t$, to prevent future leakage. For neural architectures, we additionally ensure the receptive field exceeds the longest *CE* temporal span, so the model can observe complete patterns. Baselines fall into three categories:

- **Standard end-to-end baselines.** These models take the sensor embeddings $\{v_t\}_{t=1}^T$ as input and are trained end-to-end using online *CE* labels. We consider causal sequence models over sensor embeddings: a *unidirectional LSTM* (Hochreiter & Schmidhuber, 1997), a *causal TCN* (Bai et al., 2018), a *causal Transformer encoder* with standard sinusoidal positional encoding, and *Mamba* (Gu & Dao, 2024).
- **Online action detection baseline.** We include *MiniROAD* (An et al., 2023), a top-ranked GRU-based model on benchmarks from the OAD community (Jiang et al., 2014; Geest et al., 2016), and the strongest publicly available method adaptable to our multimodal setting.
- **Neurosymbolic baselines.** These baselines apply the *ground-truth CE* rules, so performance is limited mainly by *AE* prediction errors. We include them as a reference point. *Neural AE + FSM* first predicts an *AE* label per window using an MLP classifier, then applies one FSM per *CE* rule (Section 4.1). We also explore a variant that propagates uncertainty, *Neural AE + ProbLog FSM*, implemented using the probabilistic programming language ProbLog (De Raedt et al., 2007), which takes softmax outputs from an *AE* classifier and represents FSM states as probability distributions. When the probability of an accepting state, one that corresponds to a satisfied *CE* pattern, exceeds a preset threshold, the *ProbLog FSM* is reset to its initial state.

**Implementation Details.** All models are optimized using AdamW and trained with focal loss; early stopping is based on validation loss. We report mean performance over 10 random seeds. Full baseline configurations, `NAROCE` configurations, and training details are provided in Appendix E.

**Evaluation Metrics.** We report two aggregate metrics:
(1) *Macro F1 (F1)*: For each *CE* class ($e_1$–$e_{10}$), we threshold its predicted probability at 0.5 to obtain binary predictions at each window and compute F1; we then report the unweighted average over classes.
(2) *Mean Average Precision (mAP)*: a threshold-free ranking metric. For each *CE* class, we compute average precision (AP) over all windows using the predicted probabilities, and report the mean AP over classes.

We use F1 as our primary metric and report mAP as a complementary threshold-free score.

## 6.1 Results and Analysis – Idealized Windowing

In idealized windowing, the decision window length matches the underlying sensor clip unit ($W = 2.5s$), so windows align exactly with the ground-truth *AE* boundaries. Starting from this aligned setting, we further introduce controlled boundary perturbations to study robustness to sensor noise and window misalignment.

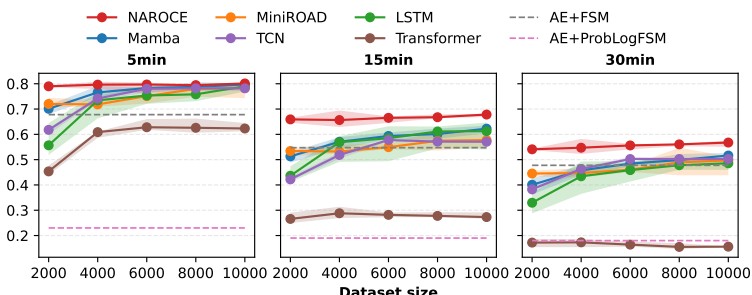

Figure 3: **Idealized aligned windowing.** Median F1 with IQR (25-75%) band. Models are trained with varying numbers of labeled sensor sequences and evaluated on 5/15/30-minute test sets.

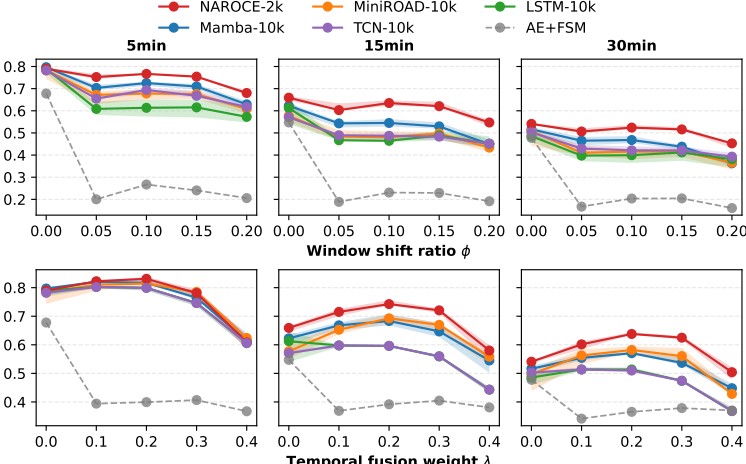

Figure 4: **Phase-shifted windowing & adjacent-window fusion.** Median F1 with IQR (25-75%) band as $\phi$ and $\lambda$ vary.

`NAROCE` **achieves the best performance with substantially less labeled sensor data.** Fig. 3 reports performance under *aligned* windowing as we vary the amount of labeled sensor training data. `NAROCE` consistently attains the highest F1 score and reaches near-peak performance with only 2k labeled sensor sequences. The advantage is most pronounced on the 15- and 30-minute test sets, where `NAROCE` maintains a clear margin over all baselines as the temporal horizon increases. Notably, the causal Transformer degrades sharply at longer horizons, likely due to the fragility of sinusoidal positional encoding when extrapolating to long contexts. The Neural AE + ProbLog FSM baseline, while conceptually appealing, also performs poorly in this setting, suggesting the need for better probabilistic FSM designs. These results support **RQ1** and **RQ2**: concept-trace pretraining substantially reduces the amount of labeled sensor data required for online CED, and improves generalization beyond the training horizon.

**Robustness to window offsets and fusion perturbations with minimal labeled sensor data.** To probe sensitivity to windowing, we evaluate two controlled variants at inference time: (i) *Phase-shifted windowing* offsets the decision windows by a shift ratio $\phi$ relative to the underlying *AE* transition boundaries. (ii) *Adjacent-window fusion* mixes neighboring windows with weight $\lambda$, $v'_t = \lambda v_t + (1 - \lambda)v_{t-1}$, as a simple temporal smoothing mechanism.

Results are shown in Fig. 4. For clarity, Mamba-10k denotes a Mamba model trained on 10k labeled sensor sequences, and `NAROCE`-2k denotes `NAROCE` trained on 2k sensor sequences (similarly for other models). Across both perturbations, `NAROCE`-2k remains better than baselines trained with the full 10k labeled set, especially on 15/30-minute horizons, supporting **RQ3**. We also observe that a small amount of fusion ($\lambda \approx 0.1$–$0.3$) improves most models, suggesting that mild temporal smoothing helps reduce sensitivity to local noise, whereas overly large fusion or shifts degrade performance. However, Neural AE + FSM degrades dramatically in both cases, highlighting that deterministic FSM is highly sensitive to *AE* misclassification caused by noise.

Table 1: **Realistic windowing.** Mean F1/mAP $\pm$ 95% CI. All test sets are cross-subject; 15/30 min additionally test longer horizons. The best and second-best scores for each column are **bold** and underlined, respectively.

| Model | Sensor | 5min | | 15min | | 30min | |
|---|---|---|---|---|---|---|---|
| | Data | F1$^\uparrow$ | mAP$^\uparrow$ | F1$^\uparrow$ | mAP$^\uparrow$ | F1$^\uparrow$ | mAP$^\uparrow$ |
| Mamba | 2,000 | .60 $\pm$ .01 | .57 $\pm$ .01 | .49 $\pm$ .02 | .47 $\pm$ .02 | .40 $\pm$ .02 | .37 $\pm$ .01 |
| | 4,000 | .66 $\pm$ .01 | .65 $\pm$ .01 | .53 $\pm$ .01 | .51 $\pm$ .01 | .43 $\pm$ .01 | .41 $\pm$ .01 |
| | 10,000 | .75 $\pm$ .01 | .76 $\pm$ .01 | .58 $\pm$ .02 | .59 $\pm$ .02 | .48 $\pm$ .02 | .48 $\pm$ .02 |
| LSTM | 2,000 | .50 $\pm$ .03 | .48 $\pm$ .02 | .36 $\pm$ .02 | .33 $\pm$ .03 | .27 $\pm$ .02 | .24 $\pm$ .03 |
| | 4,000 | .64 $\pm$ .04 | .65 $\pm$ .04 | .48 $\pm$ .03 | .49 $\pm$ .04 | .39 $\pm$ .03 | .37 $\pm$ .03 |
| | 10,000 | **.78** $\pm$ .03 | **.81** $\pm$ .04 | .57 $\pm$ .04 | .59 $\pm$ .05 | .46 $\pm$ .04 | .46 $\pm$ .04 |
| TCN | 2,000 | .56 $\pm$ .02 | .56 $\pm$ .01 | .36 $\pm$ .02 | .34 $\pm$ .01 | .31 $\pm$ .02 | .30 $\pm$ .01 |
| | 4,000 | .72 $\pm$ .01 | .74 $\pm$ .01 | .48 $\pm$ .01 | .47 $\pm$ .01 | .42 $\pm$ .01 | .41 $\pm$ .01 |
| | 10,000 | .76 $\pm$ .01 | .78 $\pm$ .01 | .53 $\pm$ .01 | .55 $\pm$ .01 | .46 $\pm$ .01 | .47 $\pm$ .01 |
| Transformer | 2,000 | .34 $\pm$ .01 | .30 $\pm$ .01 | .20 $\pm$ .01 | .16 $\pm$ .01 | .13 $\pm$ .01 | .08 $\pm$ .01 |
| | 4,000 | .44 $\pm$ .01 | .41 $\pm$ .01 | .24 $\pm$ .01 | .19 $\pm$ .01 | .14 $\pm$ .01 | .09 $\pm$ .01 |
| | 10,000 | .61 $\pm$ .02 | .61 $\pm$ .02 | .25 $\pm$ .01 | .21 $\pm$ .01 | .14 $\pm$ .01 | .09 $\pm$ .01 |
| MiniROAD | 2,000 | .63 $\pm$ .02 | .62 $\pm$ .02 | .45 $\pm$ .02 | .44 $\pm$ .04 | .36 $\pm$ .03 | .35 $\pm$ .04 |
| | 4,000 | .68 $\pm$ .01 | .69 $\pm$ .01 | .50 $\pm$ .02 | .52 $\pm$ .03 | .41 $\pm$ .02 | .41 $\pm$ .03 |
| | 10,000 | .73 $\pm$ .01 | .75 $\pm$ .01 | .58 $\pm$ .02 | .58 $\pm$ .03 | .48 $\pm$ .02 | .47 $\pm$ .02 |
| NAROCE | 2,000 | .73 $\pm$ .01 | .71 $\pm$ .01 | .56 $\pm$ .01 | .56 $\pm$ .01 | .47 $\pm$ .01 | .47 $\pm$ .01 |
| | 4,000 | .76 $\pm$ .01 | .75 $\pm$ .01 | .59 $\pm$ .02 | .60 $\pm$ .02 | .51 $\pm$ .01 | .51 $\pm$ .01 |
| | 10,000 | **.78** $\pm$ .01 | .78 $\pm$ .01 | **.62** $\pm$ .01 | **.62** $\pm$ .01 | **.53** $\pm$ .01 | **.54** $\pm$ .01 |

Table 2: F1 scores of NAROCE trained with 4k labeled sensor sequences with different detection-window sizes $W \in \{1, 2, 3, 4\}$s. The default $W = 2$s setting used in the main experiments is highlighted in gray , and the best score is in **bold**.

| Window Size | 5min | 15min | 30min |
|---|---|---|---|
| 1.0s | .72 $\pm$ .02 | .51 $\pm$ .02 | .46 $\pm$ .03 |
| 2.0s | **.76** $\pm$ .01 | .59 $\pm$ .02 | .51 $\pm$ .01 |
| 3.0s | .74 $\pm$ .01 | **.61** $\pm$ .01 | **.52** $\pm$ .01 |
| 4.0s | .73 $\pm$ .01 | .59 $\pm$ .01 | .49 $\pm$ .02 |

## 6.2 Results and Analysis - Realistic Windowing

In the realistic windowing regime, we choose decision window duration $W \neq W_{\text{clip}}$, so that it's not aligned with *AE* boundaries. We generate both the training and test splits under this setting to mimic real scenarios where decision windows cannot perfectly match *AE* durations, introducing *annotation noise/errors* when deriving window-level *CE* labels.

**NAROCE is most robust and data-efficient under realistic windowing.** Following common practice in human activity recognition (HAR), where effective window sizes typically fall within the 2–5s range (Banos et al., 2014; Wang et al., 2018), we use $W = 2.0$s as the default setting for comprehensive evaluation. Table 1 reports results under different sizes of sensor training data. Within the same data budget, NAROCE achieves the best performance on all test sets, with the largest gains on longer horizons. We also conduct a Wilcoxon Rank Sum Test (Wilcoxon, 1945) with $\alpha = 0.05$ to assess statistical significance on F1 across data budgets. Results show no significant difference between NAROCE-2k and Mamba-10k/MiniROAD-10k, the two strongest baselines, on 15/30-min test sets; NAROCE-4k significantly outperforms them on 30-min. With full budget, NAROCE-10k significantly outperforms them across all test sets. Those results further address **RQ1–RQ3** in a more realistic setting. Full statistical results are provided in Appendix F.1.

**Robustness across decision-window sizes.** Using 4k labeled sensor sequences for training, we further vary $W \in \{1, 2, 3, 4\}$s, all intentionally different from the underlying ($W_{\text{clip}} = 2.5$s) sensor-clip unit, to evaluate whether the observed performance depends on the default window-size choice. As shown in Table 2, performance is strongest for intermediate window sizes of 2–3s, while $W = 4$s remains comparable. The shorter $W = 1$s setting performs worse, likely because each window provides less evidence for identifying the underlying *AE* and substantially increases the number of reasoning steps, making long-horizon inference more susceptible to error accumulation. Conversely, larger windows may obscure transitions between adjacent *AE*s. The favorable 2–3s range is also consistent with prior HAR findings that longer windows are generally better suited to periodic activities, such as walking, than very short windows (Banos et al., 2014; Wang et al., 2018). Overall, NAROCE remains robust across a reasonable range of decision-window sizes, further supporting **RQ3**.

Table 3: Model size and FLOPs. FLOPs are reported for (i) full-sequence inference on a 5-minute trace and (ii) streaming per-step inference for a 2-second decision window in the realistic windowing setting. Models that do not support per-step inference are marked as not applicable ("/").

| Model | Params (M) | FLOPs 5-min sequence | FLOPs per 2-sec step |
|---|---|---|---|
| Mamba | 1.42 | $3.32 \times 10^7$ | $2.21 \times 10^5$ |
| Transformer | 4.30 | $5.35 \times 10^8$ | / |
| LSTM | 2.50 | $7.48 \times 10^8$ | $4.99 \times 10^6$ |
| TCN | 4.70 | $5.42 \times 10^8$ | / |
| MiniROAD | 1.62 | $4.83 \times 10^8$ | $3.21 \times 10^6$ |
| NAROCE | 2.10 | $4.98 \times 10^7$ | $3.32 \times 10^5$ |

Table 4: NAROCE trained with 2k or 4k labeled sensor data while varying the amount of CE-NAR training data (# concept traces) in the realistic windowing setting. Best scores for 2k sensor data are underlined; for 4k, **bold**. The default setting used in main experiments is highlighted in gray .

| Sensor Data | Concept Traces | 5min F1$^\uparrow$ | 5min mAP$^\uparrow$ | 15min F1$^\uparrow$ | 15min mAP$^\uparrow$ | 30min F1$^\uparrow$ | 30min mAP$^\uparrow$ |
|---|---|---|---|---|---|---|---|
| | 20k | .63 ± .02 | .61 ± .02 | .50 ± .02 | .49 ± .01 | .41 ± .01 | .38 ± .01 |
| 2,000 | 40k | .73 ± .01 | .71 ± .01 | .56 ± .01 | .56 ± .01 | .47 ± .01 | .47 ± .01 |
| | 80k | .72 ± .01 | .72 ± .01 | .57 ± .01 | .57 ± .01 | .49 ± .02 | .47 ± .01 |
| | 20k | .72 ± .01 | .70 ± .01 | .59 ± .01 | .59 ± .01 | .50 ± .02 | .49 ± .02 |
| 4,000 | 40k | **.76** ± .01 | **.75** ± .01 | .59 ± .02 | .60 ± .02 | .51 ± .01 | **.51** ± .01 |
| | 80k | .75 ± .01 | .74 ± .01 | **.61** ± .01 | **.61** ± .01 | **.52** ± .01 | **.51** ± .01 |

### 6.3 Additional Analysis

**Scalability in compute.** We report model size and FLOPs in Table 3. NAROCE adds only a small overhead over Mamba in both 5-minute full-trace inference and streaming per-step inference, while remaining 10-20× more efficient than other learning-based baselines. Transformer and TCN are not applicable for per-step inference in this setting. Overall, NAROCE is *scalable* on both axes: it reduces the labeled-sensor-data requirement and keeps streaming compute low.

**Scaling behavior of concept traces.** We study the effect of synthetic *AE*-level concept trace quantity by training the CE-NAR with 20k, 40k, and 80k examples. As shown in Table 4, increasing the number of synthetic concept traces improves both performance on the 5-minute test set and generalization to longer, out-of-distribution *CE* sequences, given the same amount of labeled sensor data. Gains are especially notable with only 2000 labeled samples. However, improvement beyond 40k concept traces is marginal, likely due to the limited size and capacity of the CE-NAR model. Overall, our findings support **Hypothesis I & II**: decoupling boosts data efficiency, and larger concept-trace pretraining improves long-horizon extrapolation and robustness

**Ablation study.** We assess the contribution of key NAROCE design choices via the ablations in Table 5: (i) removing Stage 1 and training CE-NAR and Sensor Adapter jointly from scratch; (ii) skipping the final joint finetuning of Sensor Adapter and CE-NAR after Stage 2; and (iii) replacing Focal Loss with standard multilabel cross-entropy loss in all training stages. All three ablations reduce performance. The largest drop comes from removing Stage 1, highlighting the importance of concept-trace pretraining for rule generalization. Skipping Stage 3 also hurts, indicating that joint finetuning is needed to adapt the pretrained rule prior to the target *CE*-pattern distribution. Replacing Focal Loss lowers F1, confirming the need to handle *CE* label sparsity in the loss function.

**Sensor adapter embedding reflects *AE* semantics.** As shown in Fig. 5, although trained *only* with *CE*-level supervision, the Sensor Adapter produces embeddings that form clearly separated clusters aligned with ground-truth *AE*s. This supports our **RQ4**, and provides additional interpretability for NAROCE. See Appendix F.2 for quantitative clustering metrics.

## 7 Discussion & Conclusion

**Limitations.** First, the causal Transformer baseline underperforms substantially. Although we follow Dao & Gu (2024) when selecting the Transformer encoder architecture, this design may be suboptimal for online

Table 5: Ablation study on `NAROCE` variants trained with 4k labeled sensor data. The best F1 is in **bold**.

| Arch. Variant | 5min | 15min | 30min |
|---|---|---|---|
| w/o Stage 1 Pretraining | .67 $\pm$ .01 | .53 $\pm$ .02 | .44 $\pm$ .01 |
| w/o Stage 3 Finetuning | .75 $\pm$ .01 | .57 $\pm$ .01 | .48 $\pm$ .01 |
| w/o Focal Loss | .70 $\pm$ .02 | .55 $\pm$ .02 | .47 $\pm$ .02 |
| `NAROCE` | **.76** $\pm$ .01 | **.59** $\pm$ .02 | **.51** $\pm$ .01 |

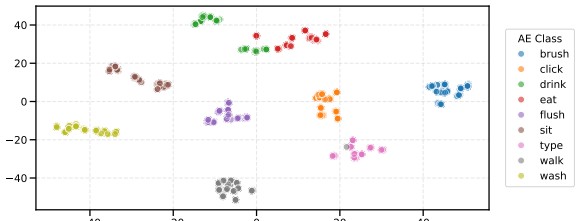

Figure 5: t-SNE visualization of Sensor Adapter's embeddings from `NAROCE_4k` on 2.5s *AE* test sensor data. Although trained solely with *CE*-level supervision, the Adapter presents semantically meaningful structure in its representation space.

CED, and its positional encoding can be fragile when generalizing to long contexts. Second, standard Transformer and TCN do not support efficient per-step streaming inference; exploring stateful or streaming designs (Gulati et al., 2020) is an important direction. Third, while `NAROCE` substantially reduces annotation requirements, 2k labeled sequences still require nontrivial data collection effort ($\approx$166 hours in our setting). Stronger data augmentation or self-supervised pretraining could further reduce annotation needs toward the hundreds, easing the burden of collecting real-world *CE* sequences. Finally, extending both the benchmark and `NAROCE` to additional sensing domains (e.g., video, WiFi), as well as collecting real-world *CE* evaluation data under privacy-preserving protocols, are important directions for future work.

**Conclusion.** We present `NAROCE`, a neural algorithmic reasoning framework for efficient and robust online CED. By decoupling *CE* rule learning from sensor-specific perception, `NAROCE` reduces reliance on large-scale labeled sensor data. We also introduce DailyOCE, a controlled and scalable multimodal testbed for online multilabel CED built from real-world sensor clips, to serve as a practical proxy for studying long-horizon CED before real-world deployment, when *CE* data are prohibitively difficult to collect and annotate. Experiments show that `NAROCE` matches or exceeds strong baselines with substantially fewer labeled sensor data and lower FLOPs, and that pretraining on large-scale, low-cost synthetic *AE*-level concept traces improves generalization to longer sequences as well as robustness to sensor noise and distribution shift.

## Broader Impact Statement

This work advances methods and benchmarks for online complex event detection from multimodal sensor streams. Potential positive impacts include more reliable monitoring and assistance in smart-health and safety applications, especially when labeled data are limited. However, models that infer high-level routines from sensor data may raise privacy concerns or be misused for surveillance. We therefore emphasize that deployments should follow informed consent, data minimization, and secure on-device or privacy-preserving processing when possible. Our benchmark is simulator-based and intended for research. However, using LLMs to synthesize simulator programs may introduce LLM-related biases and other ethical concerns. Any real-world deployment should be evaluated for bias, robustness, and safety in the target population and environment.

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

## Appendix

## A Complex Event Dataset Classes

### A.1 Complex Event Patterns

Table 6: Category of Complex Event Patterns.

| $CE$ Category | Features | Examples |
|---|---|---|
| **Sequential Patterns** | | |
| - *Relaxed* | Key $AE$s must be in order, may contain unrelated $AE$s in between | $A \rightarrow u^* \rightarrow B \rightarrow u^* \rightarrow C$, where $u$ represents user-defined unrelated $AE$s.† |
| **Temporal Patterns** | | |
| - *Duration Based* | Count the time for specific $AE$(s) | "Wash hands continuously for at least 20 seconds." "Inadequate brushing teeth that lasts less than 2 minutes, allowing a 10-second grace period in case brushing stops temporarily." |
| - *Timing Relationship* | Relative timing between different $AE$s, such as $min$, $max$ timing constraints | "After washing hands, eat within 2 minutes." |
| **Repetition Patterns** | | |
| - *Frequency Based* | Count the occurrences of specific $AE$(s) over time constraints. | "Click the mouse 5 times within 10 seconds." |
| - *Contextual Count* | Count the occurrences of specific $AE$(s) over timing related to other $AE$(s). | "After eating, wait for at least 10 minutes to work." |
| **Combination Patterns** | Sequential + Temporal Patterns | "Use Restroom $\rightarrow$ Wash (20s) $\rightarrow$ Work", (After using the restroom, ensure hands are washed for at least 20 seconds consecutively before returning to work.) |

**Notes:** †for example, the unrelated $AE$ $u$ can be any $AE$ other than the *key $AE$*s $= \{A, B, C\}$. $u^*$ means we allow for zero or more unrelated $AE$, $u$, in sequence.

### A.2 Complex Event Definitions

The 10 $CE$ classes of interest are defined as follows.

Table 7: Complex Event Classes and Definitions

| Complex Events | Definitions | Category |
|---|---|---|
| Default ($e_0$) | When no complex events of interest take place. | |
| Workspace sanitary protocol violation ($e_1$) | A violation occurs if a person starts working (click or type) without 20 seconds of consecutive handwashing after using the restroom. Upon violation, trigger an alert and reset the system. | Sequential + Temporal |
| Sanitary eating habit violation ($e_2$) | Hands are not cleaned if no 20-second consecutive handwashing occurs within 2 minutes before a meal session. A meal session starts when eating or drinking begins and ends when any activity other than eat, drink, or sit is detected. | Sequential + Temporal |
| Inadequate brushing time ($e_3$) | Brushing teeth for less than 2 minutes. If brushing stops, wait for 10 seconds; otherwise, report violation and reset the system. | Temporal (Relative + Duration) |
| Routine sequence ($e_4$) | brush $\rightarrow u^* \rightarrow$ eat $\rightarrow u^* \rightarrow$ drink $\mid$ brush $\rightarrow u^* \rightarrow$ drink $\rightarrow u^* \rightarrow$ eat, where $u = A \setminus \{$brush, eat, drink$\}$†. | Sequential - Relaxed |
| Start working and then take a break ($e_5$) | sit $\rightarrow u^* \rightarrow$ type/click $\rightarrow v^* \rightarrow$ walk, where $u = A \setminus \{$sit, type, click, walk$\}$, and $v = A \setminus \{$type, click, walk$\}$†. | Sequential - Relaxed |
| Sufficient washing reminder ($e_6$) | When washing lasts for 30 seconds consecutively. | Temporal - Duration |
| Adequate brushing time ($e_7$) | When brushing lasts a total of 2 minutes. The timer pauses if brushing stops but resumes if brushing restarts. Once the 2-minute threshold is reached, the event is reported, and the timer resets. | Temporal (Relative + Duration) |
| Post-meal rest ($e_8$) | After eating, wait for at least 3 minutes to work. | Temporal - Relative |
| Active typing session ($e_9$) | The event occurs if at least 3 typing sessions (start typing, stop typing) happen within 60 seconds of the first session's start. | Repetition - Frequency |
| Focused work start ($e_{10}$) | The event is triggered by sitting after being seated, as long as no walking occurs during this time. The event is reported after exactly 5 clicks after sitting and before walking. | Repetition - Contextual |

**Notes:** †Here $A$ represents the set of all *atomic events*.

# B  LLM-based Synthesizer

## B.1  The Prompt Template

```
1  You are a simulator that mimics daily human activities. You output sequences of activities
       as live streaming. At each window of 2.5-second, you generate a current activity label,
       which represents the activity that happens during this 2.5-second time window. Here's
       an example output of a live-streaming list of activities:
2
3  ['walk', 'sit', 'sit', 'sit', 'sit', 'flush_toilet', 'flush_toilet', 'wash', 'wash',
       'wash', 'wash', 'wash', 'wash', 'walk', 'walk', 'walk', 'walk']
4
5  Write a simulator which synthesizes random activity sequences. Follow this protocol:
6  1. Design different semantic groups (e.g., hygiene, restroom, work...) that contain related
       activities. Each group should include all activities commonly associated with it in
       realistic scenarios.
7  2. Design different range of time durations for both semantic groups and the activities in
       each group.
8  3. Design the transition between semantic groups probabilistically governed by realistic
       probabilities. Some semantic group may have higher frequency while some may happen only
       once in some period of time. Also design a distribution of the initial group.
9  4. Design the sub-transitions within a semantic group using realistic probabilities, (e.g.,
       one usually 'sit' for some time before 'flush_toilet'). Also, in reality, some activity
       may appear only once in the semantic group, use a dynamic weight adjustment so that the
       probability of ot becomes 0 once it has been selected during that group session.
10 5. Double check if the transition will give us activity patterns of interest. For instance,
       for events related to some semantic group, guarantee at least one group in the sequence
       (e.g., adjust probabilities dynamically to increase the probability of selecting this
       group after a certain amount of time has passed without it).
11 6. Add a small portion of random noise or perturbation during the generation to increase
       the sequence variability.
12
13 // User-defined input starts here
```

```
14  Now, we are interested in an event related to:
15      [USER-PROVIDED EVENT DESCRIPTION]
16
17  // User-defined activity set
18  The activities you can use to synthesize the activity traces are:
19      [USER-DEFINED ACTIVITY SET]
```

Listing 1: LLM Prompt Template

## B.2 Example Prompt & Response

Below is an example prompt used to guide the LLM in generating simulator program. In total, we use 10 LLM-generated simulators for each *CE* class to synthesize synthetic *AE*-level traces, to ensure variability in each rule pattern.

```
1  You are a simulator that mimics daily human activities. You output sequences of activities
       as live streaming. At each window of 2.5-second, you generate a current activity label,
       which represents the activity that happens during this 2.5-second time window. Here's
       an example output of a live-streaming list of activities:
2
3  ['walk', 'sit', 'sit', 'sit', 'sit', 'flush_toilet', 'flush_toilet', 'wash', 'wash',
       'wash', 'wash', 'wash', 'wash', 'walk', 'walk', 'walk', 'walk']
4
5  Write a simulator which synthesizes random activity sequences. Follow this protocol:
6  1. Design different semantic groups (e.g., hygiene, restroom, work...) that contain related
       activities. Each group should include all activities commonly associated with it in
       realistic scenarios.
7  2. Design different range of time durations for both semantic groups and the activities in
       each group.
8  3. Design the transition between semantic groups probabilistically governed by realistic
       probabilities. Some semantic group may have higher frequency while some may happen only
       once in some period of time. Also design a distribution of the initial group.
9  4. Design the sub-transitions within a semantic group using realistic probabilities, (e.g.,
       one usually 'sit' for some time before 'flush_toilet'). Also, in reality, some activity
       may appear only once in the semantic group, use a dynamic weight adjustment so that the
       probability of ot becomes 0 once it has been selected during that group session.
10 5. Double check if the transition will give us activity patterns of interest. For instance,
       for events related to some semantic group, guarantee at least one group in the sequence
       (e.g., adjust probabilities dynamically to increase the probability of selecting this
       group after a certain amount of time has passed without it).
11 6. Add a small portion of random noise or perturbation during the generation to increase
       the sequence variability.
12
13 // User-defined input starts here
14 Now, we are interested in an event related to: workspace sanitary protocol. A person may or
       may not start working without washing hands for at least 20 seconds after they use the
       restroom.
15
16 // User-defined activity set
17 The activities you can use to synthesize the activity traces are:
18 ['click_mouse', 'flush_toilet', 'sit', 'type', 'walk', 'wash']
```

Listing 2: Example Prompt

```
1  def generate_activity_sequence(duration=300):
2
3      #Simulate activities based on semantic groups, durations, and transitions.
4      #Args:
5      #    duration (int): Total simulation duration in seconds.
6
7      #Returns:
8      #    list: List of activities in 2.5-second windows.
9
10     activities = []
11
```

```
12      # Semantic groups and their activities
13      semantic_groups = {
14          "restroom": {
15              "activities": ["sit", "flush_toilet", "wash"],
16              "durations": {"sit": (2.5, 30), "flush_toilet": (2.5, 10), "wash": (2.5, 40)},
17              "transitions": {"sit": ["flush_toilet"], "flush_toilet": ["wash"], "wash": []},
18          },
19          "hygiene": {
20              "activities": ["wash"],
21              "durations": {"wash": (10, 30)},
22              "transitions": {"wash": []},
23          },
24          "work": {
25              "activities": ["sit", "click_mouse", "type"],
26              "durations": {"sit": (2.5, 20), "click_mouse": (2.5, 5), "type": (2.5, 5)},
27              "transitions": {"sit": ["click_mouse", "type"], "click_mouse": ["type"],
                    "type": ["click_mouse", "sit"]},
28          },
29          "movement": {
30              "activities": ["walk"],
31              "durations": {"walk": (2.5, 30)},
32              "transitions": {"walk": []},
33          },
34      }
35
36      # Transition probabilities between semantic groups
37      group_transitions = {
38          "restroom": [("hygiene", 0.6), ("work", 0.1), ("movement", 0.3)],
39          "hygiene": [("work", 0.8), ("movement", 0.2)],
40          "work": [("restroom", 0.2), ("work", 0.6), ("movement", 0.2)],
41          "movement": [("work", 0.2), ("restroom", 0.7), ("hygiene", 0.1)],
42      }
43
44      group_durations = {"restroom": (20, 60), "hygiene": (10, 30), "work": (10, 180),
            "movement": (2.5, 50)}
45
46      # Initial group probabilities
47      initial_group_probabilities = [
48          ("restroom", 0.1),
49          ("hygiene", 0.3),
50          ("work", 0.1),
51          ("movement", 0.5),
52      ]
53
54      def select_activity(group, current_activity=None):
55          """Select the next activity in a semantic group."""
56          if current_activity:
57              transitions = semantic_groups[group]["transitions"].get(current_activity, [])
58              if transitions:
59                  return random.choice(transitions)
60          return random.choice(semantic_groups[group]["activities"])
61
62      def weighted_choice(choices):
63          """Select an option based on weighted probabilities."""
64          total = sum(weight for _, weight in choices)
65          rand = random.uniform(0, total)
66          upto = 0
67          for choice, weight in choices:
68              if upto + weight >= rand:
69                  return choice
70              upto += weight
71
72      current_time = 0
73      current_group = weighted_choice(initial_group_probabilities)
74      while current_time < duration:
75          # Select a semantic group duration
76          group_duration = random.randint(*group_durations[current_group])
77          group_end_time = current_time + group_duration
```

```
78
79         # Generate activities for the current semantic group
80         current_activity = None
81         while current_time < group_end_time and current_time < duration:
82             activity = select_activity(current_group, current_activity)
83             activity_duration =
                   random.randint(*semantic_groups[current_group]["durations"][activity])
84             activity_duration = min(activity_duration, group_end_time - current_time,
                   duration - current_time)
85
86             # Add activities for 5-second windows
87             for _ in range(activity_duration // 5):
88                 activities.append(activity)
89
90             current_time += activity_duration
91             current_activity = activity
92
93         # Transition to the next semantic group
94         current_group = weighted_choice(group_transitions[current_group])
95
96     return activities
```

Listing 3: Example simulator code generated by LLM

## C  DailyOCE Generator

### C.1  Human activity generator

Due to the complexity of *CE* patterns, each *CE* has infinitely many combinations of *AE*s over time. To generate a general distribution for complex events, we developed a stochastic *CE* human activity simulator to concatenate multimodal time-series data from real sensor clips for each *CE* pattern.

Fig. 6 illustrates the simulator used to generate stochastic *CE* sequences. It consists of multiple *Stages*, each containing a set of *Activities* that occur with different probabilities. An *Activity* is defined by a temporal sequence of *Actions*. For example, the *Activity* "*Use Restroom*" follows the sequence: *'walk' → ('wash') → 'sit' → 'flush-toilet' → ('wash') → 'walk'*, where parentheses indicate that an *Action* occurs probabilistically. The duration of each *Action* is randomly sampled within a user-defined threshold, allowing sequences to vary in length even for the same *Activity*. Transitions between *Stages* and *Activities* are also stochastic.

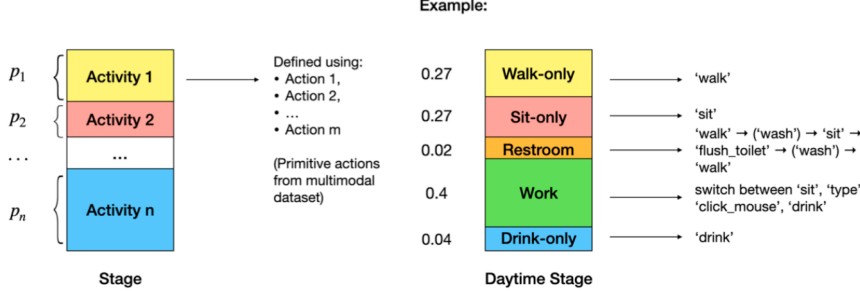

Figure 6: **Daily activity simulator.** Each *Stage* has a set of $n$ *Activities* that may happen according to a predefined distribution, where *Activity* $i$ has a probability $p_i$ of taking place in that *Stage*. Each *Activity* is defined by some temporal combinations of relevant *AE*s. For example, in *Daytime Stage Activities* "*Walk-only*", "*Sit-only*", "*Restroom*," "*Work*", and "*Drink-only*" happen with probabilities $[0.27, 0.27, 0.02, 0.4, 0.04]$ respectively. On the right side, we give some example patterns of each *Activity*.

Table 8: Definition of multimodal action classes

| Multimodal *Action* | Audio class | IMU class |
|---|---|---|
| **walk** | footsteps | walking |
| **sit** | no sound | sitting |
| **brush teeth** | brushing teeth | teeth |
| **click mouse** | mouse click | sitting |
| **drink** | drinking sipping | drinking |
| **eat** | eating | eating pasta |
| **type** | keyboard typing | typing |
| **flush toilet** | toilet flush | standing |
| **wash** | water-flowing | standing |

## C.2   Multimodal *AE* Dataset

The multimodal sensor data have 9 *Actions*, corresponding to the *AE*s. Each *Action* class is mapped to a pair of audio and IMU classes, as shown in the first column of Table. 8. The audio and IMU classes are selected from the following two datasets:

**Audio dataset.** The ESC-70 dataset is used, which is an a combination of the ESC-50 dataset Piczak (2015) and the Kitchen20 dataset Moreaux et al. (2019). The ESC-50 dataset consists of 2000 5-second labeled environmental audio recordings, with 50 different classes of natural, human, and domestic sounds. The Kitchen20 dataset collects 20 labeled kitchen-related environmental sound clips. We also self-collected 40 additional 5-second silent sound clips using MacBook Pro, 2021, for *Actions* that do not have sound. The recordings are downsampled from the original sampling rate of 44.1 kHz to 16 kHz.

**IMU dataset.** The WISDM dataset Weiss (2019) is used, containing raw accelerometer and gyroscope sensor data collected from smartphones and smartwatches, at a sampling rate of 20 Hz. It was collected from 51 test subjects as they performed 18 activities for 3 minutes each. Based on the findings in a survey paper Oluwalade et al. (2021), smartwatch data is only used for better accuracy. The original data samples are segmented into non-overlapping 2.5-second clips.

1000 multimodal data samples of 2.5 seconds per *Action* class are generated using those datasets. We sample one 2.5-second sensor clip for every *Action*, and concatenate clips according to the *AE* patterns of the generated stochastic *CE* sequences.

## D   Pretrained Feature Encoder

Fig. 7 illustrates the architecture of our pretrained feature encoder. Since the benchmark is multimodal, containing both IMU and audio streams, the encoder performs early fusion to produce a single fused embedding for each sensor window of length $W$.

### D.1   Early Fusion Model

Multimodal fusion strategies typically fall into early, late, or hybrid paradigms Baltrusaitis et al. (2017). We adopt *early fusion*, combining modalities immediately after feature extraction (Fig. 7).

Given an audio and IMU window of length $W$, we extract modality-specific embeddings using pretrained encoders: BEATs Chen et al. (2022b) for audio and LIMU-BERT Xu et al. (2021a) for IMU. Each embedding stream is passed through a GRU, projecting to 128 dimensions. We then concatenate the two 128-d embeddings into a 256-d vector and apply a fusion layer to obtain a joint 128-d representation. During pretraining, we attach a one-layer MLP *AE* classifier head. At inference, we discard the classifier head and use the remaining network as the *pretrained feature extractor*, whose window embeddings are fed into downstream complex event detectors.

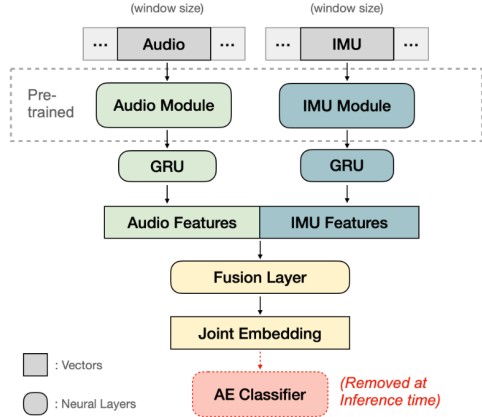

Figure 7: **Overview of the multimodal fusion module.** In the figure, a rectangular block represents data or an embedding vector, while a rounded corner rectangular block represents a neural network. The dashed-lined block indicates that it is omitted during the inference phase.

## D.2  Training

The BEATs model used for the audio module is already pre-trained, while the LIMU-Bert model for IMU module is pre-trained on large datasets. We freeze the parameters of both models and focus on training the other components of the multimodal fusion module. For training and testing the fusion module, we utilize the multimodal *AE* dataset introduced in Table. C.2. The training process minimize the cross-entropy loss:

$$\min_{\theta} L_m = -\min_{\theta} \frac{1}{N} \left( \sum_{i=1}^{N} c_i \cdot \log\left(\hat{c}_i\right) \right), \text{where } \hat{c}_i = m_\theta(\mathbf{d_i}) \tag{7}$$

where $\mathbf{d}$ is the multimodal sensor data of window size $W$, $\hat{c}_i$ is the predicted label of the *AE* in that window, $c_i$ is the corresponding ground truth label, and $N$ represents number of samples. Here we use 1000 samples.

## E  Experiment Implementations

### E.1  Baseline Model Details

**Standard End-to-end Baselines:**

- ***Unidirectional LSTM***: 5 LSTM layers with a hidden dimension of 256.

- ***Causal TCN***: We use a temporal convolutional network with causal convolutions to prevent future leakage. The network has 9 convolutional levels with channel sizes $\{128, 128, 256, 256, 256, 256, 256, 128, 128\}$ and kernel size $k = 3$. Following the standard TCN design, dilation grows exponentially across levels, and the largest dilation is $d_{\max} = 256$. The resulting receptive field is RF $= 1 + (k-1)\sum_{\ell=0}^{8} d_\ell = 1 + 2(2^9 - 1) = 1023$ windows, i.e., $1023 \times W = 1023 \times 2.5\text{s} = 2557.5\text{s} \approx 42.6$ minutes, which is larger than the longest 30-minute *CE* test case.

- ***Causal Transformer Encoder***: with a triangular attention mask to restrict the model's self-attention to previous timestamps only, excluding information from future timestamps. The causal transformer encoder uses 6 encoder layers with a hidden dimension 128, multi-head attention with 8 heads, and positional encoding Vaswani et al. (2017).

- ***Mamba***: We use a Mamba model with 12 SSM blocks. We made this design choice because the original Mamba paper Gu & Dao (2024) states that two Mamba blocks are equivalent to one transformer layer.

**Neurosymbolic Baselines.** We train an MLP $AE$ classifier on $2.5\,\text{s}$ multimodal $AE$ clips, achieving 92% accuracy on the $AE$ test set.

- **Neural AE + FSM** uses one-hot $AE$ predictions from the classifier.
- **Neural AE + ProbLog FSM** uses the classifier softmax probabilities. The accepting-state threshold is set to $1/|\mathcal{S}|$, where $|\mathcal{S}|$ is the number of FSM states.

**Online Action Detection (OAD) Baseline:** While OAD addresses a different task than online CED, its methods for long-term dependency modeling are relevant. Many recent approaches Wang et al. (2023); Chen et al. (2022a) rely on vision-specific modules (e.g., spatial attention) or task-specific components (e.g., start/end detectors, label smoothing) that are incompatible with online CED. GateHUB Chen et al. (2022a) is promising but not open-source.

We adopt MiniROAD An et al. (2023), a recent top-performing GRU-based model on standard OAD benchmarks Jiang et al. (2014); Geest et al. (2016), and the strongest publicly available option adaptable to our setup. MiniROAD identifies that RNNs underperform in long-term OAD due to a mismatch between training (short, reset clips) and inference (continuous streams). To address this, it applies a non-uniform loss weighting strategy that supervises more the final time step of each clip, aligning training behavior with inference conditions.

We adapt MiniROAD from frame-level video inputs to window-level IMU+audio segments by replacing its original visual encoder with our Pretrained Feature Encoder. While MiniROAD uses a multi-label classification loss, we replace it with our Focal Loss tailored for the online CED task, while retaining their non-uniform loss weighting strategy. We use a 4-layer MiniROAD model with a hidden dimension of 256.

### E.2 `NAROCE` Model Details

For CE-NAR, we use a Mamba model with 12 SSM blocks and hidden dimension 128. For Sensor Adapter, we use a smaller Mamba model with 6 SSM blocks and hidden dimension 128. Other neural architectures may be used, though not explored in this work.

For Stage 1 CE-NAR pretraining, all synthetic $AE$-level concept traces produced by the LLM-based Synthesizer span 5 minutes. Each $AE$ in a trace corresponds to one system window of length $W$, yielding a sequence of $T = 5\,\text{min}/W$ $AE$ tokens per trace. We generate 40,000 traces for training, 4,000 for validation, and 4,000 for testing.

### E.3 Training Details

All experiments run on one NVIDIA H100 GPU and use AdamW. For neural baselines, we use learning rate $1 \times 10^{-3}$, weight decay 0.1, and batch size 256. For `NAROCE`, we use stage-specific learning rates: $5 \times 10^{-4}$ for Stage 1 (CE-NAR pretraining), $5 \times 10^{-3}$ for Stage 2 (Sensor Adapter training), and $5 \times 10^{-5}$ for Stage 3 (joint finetuning). We use dropout 0.1 in the idealized setting to reduce overfitting and dropout 0 in the realistic setting. The Sensor Adapter is a 3-layer Mamba module in the idealized setting to reduce overfitting. Baseline architectures are kept fixed across windowing regimes: after an architecture search, we use the same selected configurations in both settings, as we found baselines consistently benefit from larger model sizes.

We train with multilabel Focal Loss to address severe label imbalance. Early stopping is based on validation loss with a fixed patience. We cap training at $2 \times 10^8$ samples for all models and report results over 10 random seeds.

### E.4 Training Details

All experiments run on one NVIDIA H100 GPU and use AdamW.

**Optimization Hyperparameters.** For neural baselines, we use learning rate $1 \times 10^{-3}$, weight decay 0.1, and batch size 256. For `NAROCE`, we use stage-specific learning rates: $5 \times 10^{-4}$ for Stage 1 (CE-NAR pretraining), $5 \times 10^{-3}$ for Stage 2 (Sensor Adapter training), and $5 \times 10^{-5}$ for Stage 3 (joint finetuning). We use dropout 0.1 in the idealized setting to reduce overfitting and dropout 0 in the realistic setting. In the

idealized setting, we use a 3-layer Mamba Sensor Adapter to reduce overfitting. Baseline architectures are kept fixed across windowing regimes: after an architecture search, we use the same selected configurations in both settings, as we found baselines consistently benefit from larger model sizes.

We train with multilabel Focal Loss to address severe label imbalance. Early stopping is based on validation loss with a fixed patience. We report results over 10 random seeds.

**Training Schedule.** Stage 1 (CE-NAR pretraining) is trained for 5,000 epochs. Stage 2 (Sensor Adapter training) is trained for a total of 200,000,000 sensor-sample iterations (i.e., the total number of per-sample update steps accumulated over repeated passes through the dataset). All neural baselines are trained with the same budget of 200,000,000 sensor-sample iterations. Stage 3 (joint finetuning) is trained for 10 epochs.

## F   More results on `NAROCE`

### F.1   Detailed Results - Wilcoxon Statistical Test

**Comparison between `NAROCE_2k` and `mamba_10k`.** We conduct two one-sided Wilcoxon rank-sum tests to evaluate the null hypotheses: (1) $H_0$: `NAROCE_2k` is worse than mamba_10k and (2) $H_0$: `NAROCE_2k` is better than mamba_10k. For *CE 15-min*, the resulting p-values are 0.9894 (testing `NAROCE_2k` > mamba_10k) and 0.083 (testing mamba_10k > `NAROCE_2k`); for *CE 30-min*, they are 0.8793 and 0.1365, respectively. None of these p-values is significant enough to reject either hypothesis, so we conclude that there is **no significant difference** between `NAROCE_2k` and mamba_10k on the 15- and 30-minute *CE* test sets.

**Comparison between `NAROCE_2k` and `miniroad_10k`.** We conduct two one-sided Wilcoxon rank-sum tests to evaluate the null hypotheses: (1) $H_0$: `NAROCE_2k` is worse than miniroad_10k and (2) $H_0$: `NAROCE_2k` is better than miniroad_10k. For *CE 15-min*, the resulting p-values are 0.9894 (testing `NAROCE_2k` > miniroad_10k) and 0.0829 (testing miniroad_10k > `NAROCE_2k`); for *CE 30-min*, they are 0.9071 and 0.1061, respectively. None of these p-values is significant enough to reject either hypothesis, so we conclude that there is **no significant difference** between `NAROCE_2k` and miniroad_10k on the 15- and 30-minute *CE* test sets.

**Comparison between `NAROCE_4k` and `mamba_10k`.** We conduct a one-sided Wilcoxon rank-sum (Mann–Whitney $U$) test to evaluate the null hypothesis $H_0$: `NAROCE_4k` is worse than mamba_10k on the *CE 30-min* test set. The resulting p-value is 0.0320, which is significant at the 0.05 level. Thus, `NAROCE_4k` is significantly better than mamba_10k on *CE 30-min*.

**Comparison between `NAROCE_4k` and `miniroad_10k`.** We conduct a one-sided Wilcoxon rank-sum (Mann–Whitney $U$) test to evaluate the null hypothesis $H_0$: `NAROCE_4k` is worse than miniroad_10k on the *CE 30-min* test set. The resulting p-value is 0.0156, which is significant at the 0.05 level. Thus, `NAROCE_4k` is significantly better than miniroad_10k on *CE 30-min*.

**Comparison between `NAROCE_10k` and `mamba_10k`.** We conduct one-sided Wilcoxon rank-sum (Mann–Whitney $U$) tests to evaluate the null hypothesis $H_0$: `NAROCE_10k` is worse than mamba_10k. The resulting p-values are $9.13 \times 10^{-5}$, 0.00229, and 0.000658 for *CE 5-min*, *CE 15-min*, and *CE 30-min*, respectively. Since all p-values are significant at the 0.05 level, we conclude that `NAROCE_10k` significantly outperforms mamba_10k across all three horizons.

**Comparison between `NAROCE_10k` and `miniroad_10k`.** We conduct one-sided Wilcoxon rank-sum (Mann–Whitney $U$) tests to evaluate the null hypothesis $H_0$: `NAROCE_10k` is worse than miniroad_10k. The resulting p-values are $9.13 \times 10^{-5}$, 0.00141, and 0.000220 for *CE 5-min*, *CE 15-min*, and *CE 30-min*, respectively. Since all p-values are significant at the 0.05 level, we conclude that `NAROCE_10k` significantly outperforms miniroad_10k across all three horizons.

## F.2 *AE*-structure of Sensor Adapter Embedding

t-SNE visualization of the latent embedding produced by the Sensor Adapter of `NAROCE_2k` and `NAROCE_4k` in idealized setting on train and test sets are shown in Fig. 8. Clear separation of classes indicates the *AE*-level structure is captured by the Sensor Adapter despite no explicit *AE* supervision during training.

We report Adjusted Rand Index (ARI), Normalized Mutual Information (NMI), and linear probe accuracy in Table 9. High scores across all metrics indicate that the Sensor Adapter effectively reflects *AE*-level structure in its latent space, despite *CE*-only supervision. Notably, the model trained with more annotated *CE* sensor data (`NAROCE_4k`) achieves higher performance, demonstrating better generalization, especially on unseen *AE* test sensor data.

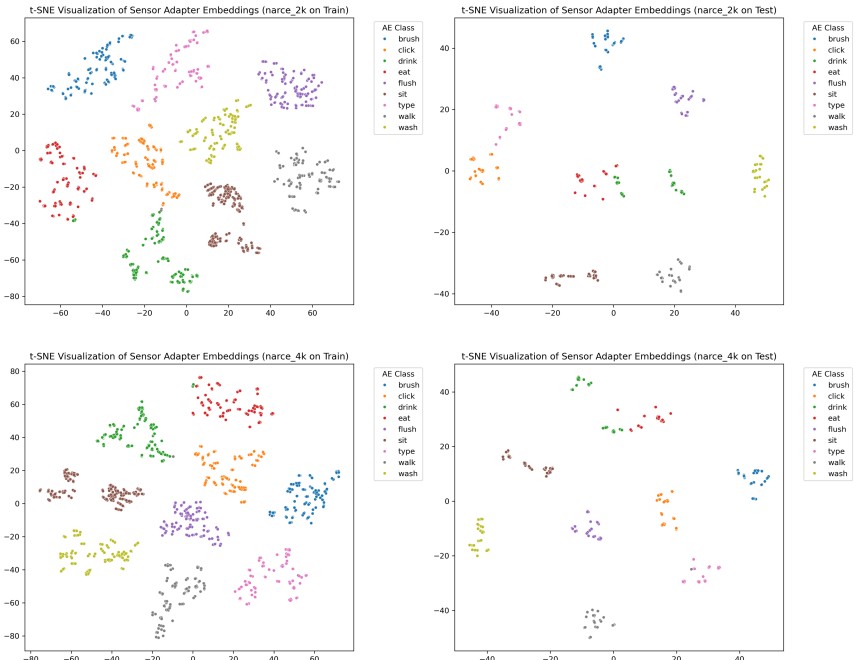

Figure 8: t-SNE visualization of the latent embedding produced by the Sensor Adapter of `NAROCE_2k` and `NAROCE_4k` on *AE* Train and Test sensor data. Points are colored by *AE* ground truth classes.

Table 9: Quantitative analysis of *AE*-like structure in Sensor Adapter embeddings, evaluated across `NAROCE_2k` and `NAROCE_4k` on Train and Test sets.

| Model | Dataset | ARI | NMI | Linear Probe Accuracy |
|---|---|---|---|---|
| NAROCE_2k | Train | 0.976 | 0.977 | 0.994 |
| NAROCE_2k | Test | 0.900 | 0.935 | 0.956 |
| NAROCE_4k | Train | 0.985 | 0.985 | 0.997 |
| NAROCE_4k | Test | 0.903 | 0.943 | 0.967 |

## G FSM Examples

Examples FSM Codes for $e_1$, $e_2$ and $e_3$ defined in Table 7. Those implementations utilize an Extended Finite State Machine (eFSM) instead of a standard FSM to improve efficiency in counting tasks and simplify state logic. eFSMs enhance readability by incorporating variables and conditions for state transitions, making

them easier to understand and manage. While any eFSM can still be represented as a standard FSM, using an eFSM allows for a more concise and structured approach to state management for easy interpretation.

---

**Algorithm 1** State Machine for Complex Event 1 Detection

---

**Require:** An activity input $x$ at time $t$
**Ensure:** A complex event label $y \in \{0, 1\}$; Returns 1 if the event of interest is detected at $t$, 0 otherwise
1: $y \leftarrow 0$
2: $state \leftarrow 0$         ▷ The initial state
3: $wash\_counter \leftarrow 0$         ▷ Counter for continuous wash activities after restroom use
4: **function** STATE_MACHINE_1($x$)
5:      **if** $state = 0$ **then**
6:         **if** $x = flush\_toilet$ **then**
7:            $state \leftarrow 1$         ▷ Transition to *After restroom use* state
8:            $wash\_counter \leftarrow 0$         ▷ Reset wash counter
9:         **end if**
10:      **else if** $state = 1$ **then**
11:         **if** $x = wash$ **then**
12:            $wash\_counter \leftarrow wash\_counter + 1$
13:            ▷ Increment wash counter
14:            **if** $wash\_counter \geq 20$ **then**
15:               $state \leftarrow 0$         ▷ Reset to initial state after sufficient washing (20 seconds)
16:            **end if**
17:         **else if** $x = click\_mouse$ or $x = type$ **then**         ▷ Working behavior
18:            **if** $wash\_counter < 20$ **then**
19:               $y \leftarrow 1$         ▷ Event detected
20:            **end if**
21:            $state \leftarrow 0$         ▷ Reset state
22:         **else**
23:            $wash\_counter \leftarrow 0$         ▷ Reset wash counter for other activities
24:         **end if**
25:      **end if**
26:      **return** $y$
27: **end function**

---

**Algorithm 2** State Machine for Complex Event 2 Detection

---

**Require:** An activity input $x$ at time $t$
**Ensure:** A complex event label $y \in \{0, 2\}$; Returns 2 if the event of interest is detected at $t$, 0 otherwise
1: $y \leftarrow 0$
2: $state \leftarrow 0$         ▷ The initial state
3: $wash\_count \leftarrow 0$         ▷ Counter for continuous wash activities
4: $time\_since\_wash \leftarrow \infty$         ▷ Time since last wash
5: $is\_meal \leftarrow False$
6: **if** $x = eat$ or $x = drink$ **then**
7:      $is\_meal \leftarrow True$         ▷ Check if the input is a meal activity
8: **end if**
9:
10: **function** STATE_MACHINE_2($x$)
11:      **if** $state = 0$ **then**
12:         $wash\_count \leftarrow 0$
13:         **if** $x = wash$ **then**
14:            $state \leftarrow 1, wash\_count \leftarrow 1$
15:         **else if** $is\_meal$ **then**
16:            $state \leftarrow 3, y \leftarrow 2$         ▷ Event detected
17:         **end if**
18:      **else if** $state = 1$ **then**         ▷ *Washing hands* state, hands are not clean yet
19:         **if** $x = wash$ **then**
20:            $wash\_count \leftarrow wash\_count + 1$
21:            **if** $wash\_count \geq 20$ **then**         ▷ Washing for sufficient time (20 seconds)
22:               $state \leftarrow 2$         ▷ Move to *Clean hands* state
23:               $time\_since\_wash \leftarrow 0$
24:            **end if**
25:         **else if** $is\_meal$ **then**
26:            $state \leftarrow 3, wash\_count \leftarrow 0, y \leftarrow 2$         ▷ Event detected

```
27:        else
28:            state ← 0, wash_count ← 0
29:        end if
30:    else if state = 2 then                                              ▷ Clean hands state
31:        if is_meal then
32:            state ← 3                                          ▷ Stay in Clean hands state during meal
33:        else if x ∈ {brush_teeth, click_mouse, flush_toilet, type} then
34:            state ← 0                                   ▷ Need to wash hands again after touching things
35:        else if x = wash then
36:            time_since_wash ← 0                              ▷ Reset timer, but stay in Clean hands state
37:        else
38:            time_since_wash ← time_since_wash + 1
39:        end if
40:        if time_since_wash > 120 then
41:            state ← 0                              ▷ More than 2 minutes passed since last wash, need
                                                                                     to rewash hands
42:        end if
43:    else if state = 3 then                                     ▷ Having meals state, stop the timer
44:        if is_meal or x = sit then
45:            continue                                        ▷ Stay in the Having meals state
46:        else if x ∈ {brush_teeth, click_mouse, flush_toilet, type} then
47:            state ← 0                               ▷ Need to wash hands again after touching things
48:        else if x = wash then
49:            time_since_wash ← 0
50:            if wash_count ≥ 20 then
51:                state ← 2                                       ▷ Go back to Clean hands state
52:            else
53:                state ← 1
54:            end if
55:        else
56:            if wash_count ≥ 20 then                             ▷ Go back to Clean hands state
57:                time_since_wash ← time_since_wash + 1
58:                state ← 2
59:            else
60:                state ← 0
61:            end if
62:        end if
63:    end if
64:    return y
65: end function
```

---

**Algorithm 3** State Machine for Complex Event 3 Detection

---

**Require:** An activity input $x$ at time $t$
**Ensure:** A complex event label $y \in \{0, 3\}$; Returns 3 if the event of interest is detected at $t$, 0 otherwise

```
1: y ← 0
2: state ← 0                                                          ▷ The initial state
3: brush_counter ← 0                                           ▷ Counter for total brushing time
4: time_since_brush ← 0                                              ▷ Time since last brush
5:
6: function STATE_MACHINE_3(x)
7:     if state = 0 then                                                 ▷ Initial state
8:         if x = brush_teeth then
9:             state ← 1
10:            brush_counter ← brush_counter + 1
11:        end if
12:    else if state = 1 then                                          ▷ Brushing teeth state
13:        if x = brush_teeth then
14:            brush_counter ← brush_counter + 1
15:        else
16:            state ← 2
17:            time_since_brush ← time_since_brush + 1
18:        end if
19:    else if state = 2 then                                    ▷ Wait for further brushing state
20:        if x = brush_teeth then
21:            state ← 1                                          ▷ Return to Brushing teeth state
22:            brush_counter ← brush_counter + 1
23:            time_since_brush ← 0                                 ▷ Reset counter for other actions
```

```
24:        else
25:            time_since_brush ← time_since_brush + 1
26:            temp ← brush_counter                                    ▷ Record brushing time
27:            if timesincebrush > 2 then
28:                state ← 0                                           ▷ Return to initial state
29:                brush_counter ← 0
30:                time_since_brush ← 0
31:                if temp < 120 then                      ▷ Check if brushing was less than 2 minutes
32:                    y ← 3                                           ▷ Event detected
33:                end if
34:            end if
35:        end if
36:    end if
37:    return y
38: end function
```

# H    Realism and Structural Diversity of the Dataset

***CE* Class Distribution.** Table 10 shows the distribution of each CE across the training set, 5-minute test set, and the OOD 15-minute and 30-minute test sets. We calculate the occurrence rate of each *CE* across all data samples.

Table 10: Percentage of samples containing each *CE* class across datasets.

| Dataset | $e_0$ | $e_1$ | $e_2$ | $e_3$ | $e_4$ | $e_5$ | $e_6$ | $e_7$ | $e_8$ | $e_9$ | $e_{10}$ | Only $e_0$ |
|---|---|---|---|---|---|---|---|---|---|---|---|---|
| Train | 100 | 16.5 | 27.9 | 19.5 | 10.8 | 18.8 | 20.8 | 13.8 | 7.3 | 10.5 | 13.9 | 9.1 |
| Test-5min | 100 | 16.9 | 26.5 | 19.9 | 11.0 | 18.9 | 20.9 | 13.5 | 6.4 | 10.2 | 14.1 | 9.7 |
| Test-15min | 100 | 10.2 | 24.0 | 14.7 | 20.2 | 13.0 | 62.9 | 18.7 | 16.7 | 28.9 | 40.3 | 1.4 |
| Test-30min | 100 | 10.7 | 28.5 | 15.2 | 19.4 | 18.1 | 71.8 | 18.3 | 20.2 | 42.7 | 50.9 | 0.2 |

**Temporal Span Diversity.** Fig. 9 highlights the diversity of *CE* durations across datasets. While the training and 5-minute test sets have similar distributions of *CE* spans, the OOD test sets (15-minute and 30-minute sequences) introduce longer and more variable event durations, creating a meaningful generalization challenge for detection models.

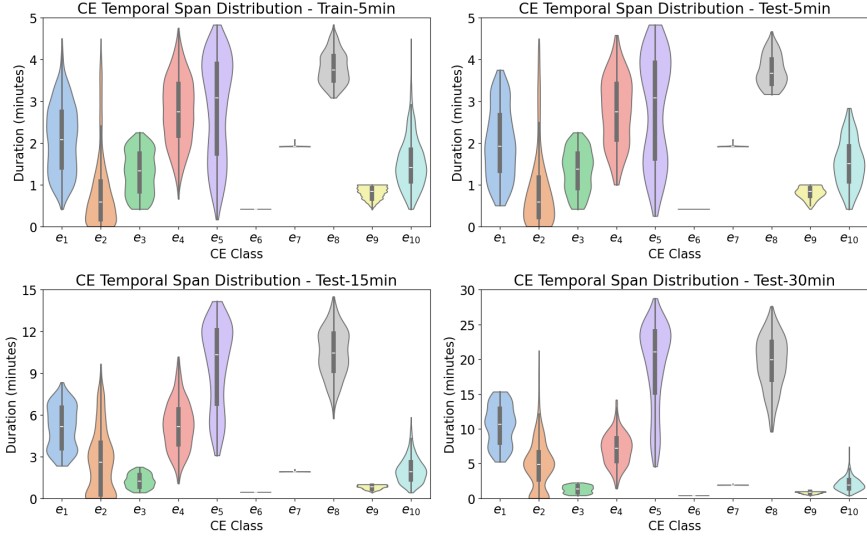

Figure 9: Distribution of the temporal span of each *CE* across datasets.

**CE Overlap Statistics.** Table 11 presents detailed statistics on event overlaps, and Fig. 10 visualizes the pairwise distribution of overlapping *CE*s. These insights illustrate the complexity of the dataset's temporal dependencies.

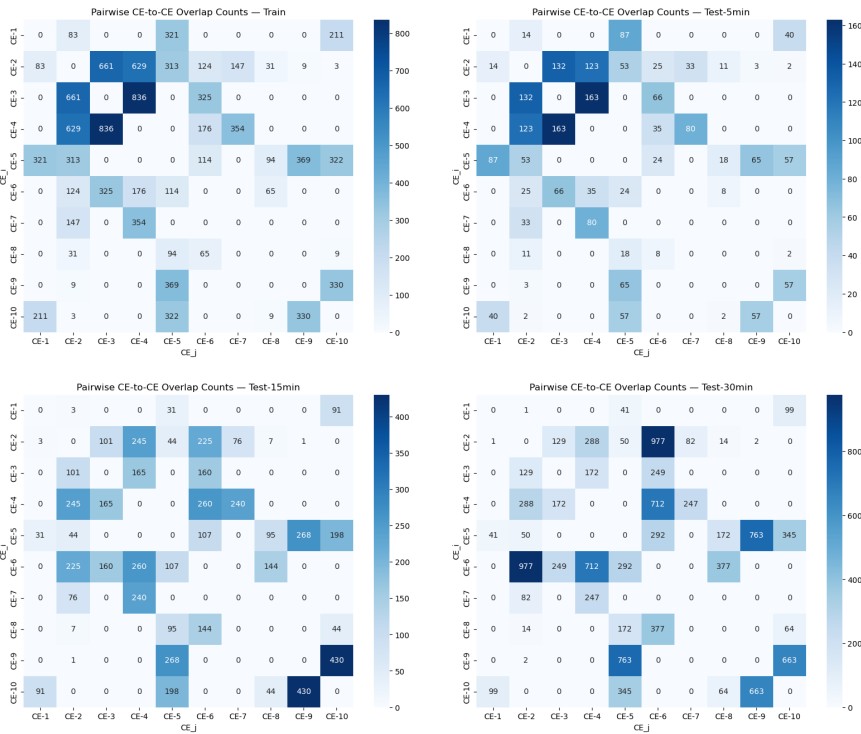

Figure 10: Pairwise CE-to-CE overlap heatmaps across datasets. Each cell CE_i, CE_j, $i \neq j$, indicates how many samples contain at least one CE_i and CE_j whose temporal spans overlap.

(a) Train

| CE | % Overlap | Min | Max | Max Inst. |
|---|---|---|---|---|
| $e_1$ | 36.2 | 0 | 3 | 3 |
| $e_2$ | 45.4 | 0 | 3 | 3 |
| $e_3$ | 62.5 | 0 | 2 | 2 |
| $e_4$ | 100.0 | 1 | 2 | 3 |
| $e_5$ | 69.1 | 0 | 4 | 5 |
| $e_6$ | 29.6 | 0 | 2 | 2 |
| $e_7$ | 32.4 | 0 | 2 | 2 |
| $e_8$ | 25.6 | 0 | 2 | 2 |
| $e_9$ | 57.0 | 0 | 2 | 3 |
| $e_{10}$ | 62.5 | 0 | 3 | 4 |

(b) Test-5min

| CE | % Overlap | Min | Max | Max Inst. |
|---|---|---|---|---|
| $e_1$ | 37.5 | 0 | 3 | 4 |
| $e_2$ | 44.2 | 0 | 3 | 3 |
| $e_3$ | 65.7 | 0 | 2 | 2 |
| $e_4$ | 100.0 | 1 | 2 | 2 |
| $e_5$ | 67.2 | 0 | 4 | 5 |
| $e_6$ | 29.5 | 0 | 2 | 2 |
| $e_7$ | 33.6 | 0 | 2 | 2 |
| $e_8$ | 24.5 | 0 | 2 | 2 |
| $e_9$ | 54.0 | 0 | 2 | 2 |
| $e_{10}$ | 58.2 | 0 | 3 | 3 |

(c) Test-15min

| CE | % Overlap | Min | Max | Max Inst. |
|---|---|---|---|---|
| $e_1$ | 46.7 | 0 | 3 | 3 |
| $e_2$ | 62.4 | 0 | 3 | 5 |
| $e_3$ | 84.2 | 0 | 3 | 3 |
| $e_4$ | 100.0 | 1 | 3 | 4 |
| $e_5$ | 98.2 | 0 | 5 | 9 |
| $e_6$ | 35.0 | 0 | 3 | 3 |
| $e_7$ | 60.8 | 0 | 2 | 2 |
| $e_8$ | 63.8 | 0 | 3 | 4 |
| $e_9$ | 62.3 | 0 | 2 | 3 |
| $e_{10}$ | 58.8 | 0 | 3 | 5 |

(d) Test-30min

| CE | % Overlap | Min | Max | Max Inst. |
|---|---|---|---|---|
| $e_1$ | 63.4 | 0 | 2 | 2 |
| $e_2$ | 79.8 | 0 | 3 | 9 |
| $e_3$ | 92.6 | 0 | 3 | 3 |
| $e_4$ | 100.0 | 1 | 3 | 7 |
| $e_5$ | 100.0 | 1 | 5 | 11 |
| $e_6$ | 43.6 | 0 | 3 | 3 |
| $e_7$ | 64.9 | 0 | 2 | 2 |
| $e_8$ | 78.7 | 0 | 3 | 6 |
| $e_9$ | 56.9 | 0 | 2 | 3 |
| $e_{10}$ | 67.8 | 0 | 3 | 6 |

Figure 11: Overlap statistics for each CE across datasets. "% Overlap" shows the percentage of samples where a *CE* overlaps with others. "Min/Max" are the number of overlapping *CE* types, and "Max Inst." shows the maximum number of overlapping *CE* instances observed in a single sample.

As these tables and figures prove, although the dataset is synthetic, its structure is principled, realistic, and suitable for studying online *CE* reasoning.

