# OpenReview forum: "Scaling Online Complex Event Detection with Synthetic Supervision and Mamba-Based Neural Algorithmic Reasoning"
_TMLR — Under review for TMLR_

### Review · Reviewer_wbsk · 2026-06-30

**Summary Of Contributions:**

This paper studies online complex event detection, where a model detects high-level events defined by temporal or logical rules over streams of lower-level atomic events. The authors identify several key challenges in this setting, including long temporal dependencies, generalization beyond the training horizon, sparse CE-level supervision, and the high cognitive cost of annotating complex events. To address these challenges, they propose NAROCE, a neural algorithmic reasoning framework that decouples high-level rule reasoning from low-level sensor perception. The paper also introduces DailyOCE, a controlled online multilabel CED benchmark constructed by composing real IMU and audio clips into longer sensor traces.

**Strengths**
1. The decomposition into concept-level rule pretraining and sensor-to-reasoner adaptation is intuitive and potentially useful.
2.  The method is computationally reasonable, and the Mamba-based design is well motivated for streaming inference.
3. The paper evaluates several important dimensions: data efficiency, long-horizon extrapolation, realistic windowing, compute cost, synthetic pretraining scale, ablations, and adapter representations.

**Weaknesses**
1. Current experiments mainly support the effectiveness of NAROCE in a controlled benchmark but do not fully establish its performance in naturally collected real-world online CED scenarios.
2. The use of AE-level supervision to train the pretrained feature encoder needs clearer accounting, because the paper emphasizes CE-only supervision for downstream training.
3. Some baselines need stronger justification or improvement, particularly the weak causal Transformer baseline and the AE-plus-FSM baselines.
4. The LLM-based simulator is central to the method but under-analyzed.

**Audience:**

Yes

**Audience Explanation:**

Researchers working on long-horizon sequence modeling, online inference, and state-space models may find the application of Mamba to streaming complex-event reasoning useful. The idea of treating CE-rule execution as an algorithm over symbolic atomic-event traces and then transferring that learned procedure to noisy sensor embeddings is a natural and interesting extension of NAR ideas beyond clean symbolic inputs. Besides, the benchmark contribution may interest researchers in multimodal learning, sensor-based activity recognition, and neuro-symbolic learning.

**Broader Impact Concerns:**

The paper already includes a broader impact statement, and it correctly identifies privacy and surveillance as potential concerns.

**Claims And Evidence:**

No

**Claims Explanation:**

The claims are partially supported, but not yet supported convincingly enough.
1. DailyOCE is a controlled proxy benchmark rather than a real long-horizon CE dataset. Although it uses real IMU and audio clips, the event traces are constructed by a simulator and the CE labels are generated from explicit FSMs. This is a reasonable experimental design for controlled evaluation, but it limits the strength of claims about real-world CED.
2. The role of AE supervision is somewhat ambiguous. The paper states that downstream CED uses only CE-level supervision and that AE labels are unavailable for training the CED model. However, the frozen pretrained feature encoder is trained on AE clips segmented using the same decision window.
3. The Transformer baseline performs very poorly at long horizons, but the paper attributes this partly to sinusoidal positional encoding. This is plausible, but not sufficient. A stronger evaluation should include relative/rotary position encodings. Similarly, the AE-plus-FSM and ProbLog-FSM baselines are conceptually important but underperform dramatically. Since these baselines use the ground-truth rules, their poor performance likely reflects AE prediction error and windowing noise.
4. The paper uses GPT-4o to generate simulator programs, but there is limited evidence that the generated programs are diverse, unbiased, or necessary. It is also unclear how sensitive NAROCE is to simulator mismatch.

**Requested Changes:**

1. The authors should clarify that the current evidence supports NAROCE on a controlled simulator-based benchmark, not on naturally collected real-world long-horizon CED data.
2. Clarify the supervision used by the full pipeline.
3. The paper should distinguish more carefully between learning new CE rules and learning to execute predefined CE rules, since the event rules are predefined rather than discovered from data.

---

### Review · Reviewer_je2k · 2026-07-07

**Summary Of Contributions:**

This paper studies online multilabel complex event detection (CED), where high-level complex events (CEs) must be detected causally from streaming multimodal sensor data based on temporal, duration-based, repetition-based, or compositional rules over atomic events (AEs). The proposed framework, NAROCE, decomposes the task into CE rule reasoning and sensor-to-reasoner alignment: it pretrains a Mamba-based CE rule reasoner, CE-NAR, on synthetic AE-level concept traces, then trains a Sensor Adapter with limited CE-labeled sensor sequences, followed by light joint finetuning. The paper also introduces DailyOCE, a benchmark built by composing real IMU/audio clips and generating multilabel CE labels using expert-defined FSM rules, supporting cross-subject evaluation, long-horizon extrapolation, window perturbations, and different decision-window sizes. Experiments show that NAROCE is competitive with or better than strong baselines in data efficiency, long-horizon generalization, and robustness, especially under limited labeled sensor data. Overall, I find the paper well motivated and technically interesting. Its main strengths are the clear problem formulation, the natural decomposition between rule reasoning and sensor alignment, the systematic evaluation, and the potential community value of DailyOCE. The main issues are about making several claims more precise.

**Audience:**

Yes

**Audience Explanation:**

The paper should interest TMLR readers working on long-sequence modeling, neural algorithmic reasoning, synthetic supervision, multimodal sensor learning, online action/event detection, and neurosymbolic reasoning. Its value is not only in the performance gains, but also in the decomposition between high-level CE rule learning and low-level sensor semantic learning.

DailyOCE also has potential community value as a controlled environment for evaluating multilabel, long-horizon, rule-governed online CED methods. Although it remains simulator-based, it enables stress tests that would be difficult to cover at scale with real-world CE data.

**Broader Impact Concerns:**

The paper already discusses privacy, surveillance risks, informed consent, data minimization, and the need for real-world bias and robustness evaluation. I find this generally sufficient. One minor addition worth considering is that the proposed approach lowers the data requirement for building high-level behavior monitoring systems. This is a technical advantage, but it may also lower the barrier for misuse in surveillance-like applications.

**Claims And Evidence:**

Yes

**Claims Explanation:**

The core claim that synthetic concept-trace pretraining improves data efficiency and long-horizon generalization is generally well supported. The experiments cover different labeled data budgets, 5/15/30-minute test sequences, idealized and realistic windowing, window shifts, and different decision-window sizes. The ablation study also supports the importance of Stage 1 concept-trace pretraining.

However, a few claims should be made more precise. First, the paper emphasizes CE-level supervision and interprets the Sensor Adapter embeddings as learning AE-level semantic structure from CE supervision. Since the pretrained feature encoder is trained with AE-labeled clips, the AE clustering in the adapter output cannot be fully attributed to CE-level supervision alone. The claim should be toned down, or supported by a comparison between the feature encoder output \(v_t\) and adapter output \(f'(v_t)\).

Second, the statistical testing language should be revised: not finding a statistically significant difference is not evidence of equivalence. Third, the “10–20× fewer FLOPs” claim should specify that it does not apply to the Mamba baseline, for which NAROCE has higher FLOPs. Finally, DailyOCE is a valuable controlled proxy, but concept-trace pretraining and benchmark labeling share the same expert-defined FSM rules. The paper should therefore state more explicitly that the evidence supports rule extrapolation and sensor-to-rule alignment within a controlled rule-generated benchmark, rather than open-world generalization to unseen real-world CE semantics.

**Requested Changes:**

1. Clarify the supervision used in each part of the pipeline. The paper should distinguish synthetic AE-level concept traces, AE-labeled clips for the pretrained feature encoder, and CE-labeled sensor sequences. The current “CE-only supervision” wording is too broad unless qualified.

2. Revise the interpretation of the Sensor Adapter embedding analysis. Since the input feature encoder is trained with AE labels, the AE clustering in adapter outputs does not by itself show that AE structure emerges solely from CE-level supervision. The claim should be toned down, or supported by a comparison between \(v_t\) and \(f'(v_t)\).

3. Correct the statistical testing language. “No significant difference” should not be presented as evidence of equivalence. The authors should either run an equivalence test or state more cautiously that no statistically significant difference was detected.

4. Qualify the FLOPs claim in the abstract and conclusion. The “10–20× fewer FLOPs” statement should make clear that it does not apply to the Mamba baseline.

5. Make the scope of DailyOCE clearer. Since concept-trace pretraining and benchmark labeling share the same FSM rule definitions, the paper should explicitly state that the evidence supports rule extrapolation and sensor-to-rule alignment within a controlled rule-generated benchmark.

---

### Review · Reviewer_EgsC · 2026-07-12

**Summary Of Contributions:**

The paper studies online multilabel complex-event detection from audio and IMU streams. It proposes NAROCE, which contains a CE reasoner that generates synthetic atomic-event sequences and an adapter using CE-labeled sensor sequences. The central empirical claim is that access to synthetic rule supervision enables NAROCE to approach or exceed standard sequence models with fewer CE-labeled sensor traces and relatively low streaming inference cost.

## Strength
The problem is important and underexplored. The distinction between short-horizon perception and long-horizon temporal reasoning is useful, and online satisfaction-time detection is meaningfully different from ordinary activity recognition.

**Audience:**

Yes

**Audience Explanation:**

Pretraining a state-space reasoner on synthetic atomic-event traces generated from known temporal rules can reduce the amount of labeled sensor data needed for online complex-event detection, while maintaining relatively low streaming inference cost. The combination of synthetic supervision, Mamba-based long-sequence modeling, multimodal sensing, and online detection is relevant to researchers working on neural algorithmic reasoning, weak supervision, state-space models, neurosymbolic learning, and sensor-based activity understanding.

**Broader Impact Concerns:**

I don't have any concerns about the ethical part.

**Claims And Evidence:**

No

**Claims Explanation:**

The paper is very confusing to read, my major concerns are following:

1. **The paper does not actually eliminate rule specification; it distills known rules** The manuscript repeatedly describes CE-rule learning as the primary bottleneck and contrasts NAROCE with neurosymbolic methods that require user-defined rules. However, NAROCE itself requires an executable expert-designed FSM for every CE. These FSMs are used to produce all Stage 1 labels and all benchmark labels.
2. **The claim of CE-only supervision is contradicted by the feature-encoder training** Section 3 states that fine-grained AE annotations are unavailable and that only coarse CE labels are supplied. Section 4 similarly states that training supervision remains CE-only. However, Appendix D trains the shared feature encoder using cross-entropy on explicit ground-truth AE classes. The paper first states that 1,000 multimodal clips are generated per action class, but Equation 7 then says $N=1000$, leaving the total supervision budget ambiguous.
3. **The “long-horizon generalization” experiment is substantially confounded** The 15- and 30-minute test sets do not differ from training only in horizon or CE temporal span. Table 10 shows large class-prevalence changes. For example, $e_6$ changes from 20.8% in training to 71.8% at 30 minutes.

**Requested Changes:**

1. Work on the readability of the manuscript; the current submission is very difficult to follow.
2. Correct the CE-only-supervision claims and include the AE-label budget.
3. Report clean symbolic CE-NAR accuracy and controlled AE-corruption experiments.
4. Provide controlled horizon experiments that hold CE prevalence and event counts fixed.